# PARALLEL SCALING LAW: UNVEILING REASONING GENERALIZATION THROUGH A CROSS-LINGUISTIC PERSPECTIVE

## ABSTRACT

Recent advancements in Reinforcement Post-Training (RPT) have significantly enhanced the capabilities of Large Reasoning Models (LRMs), sparking increased interest in the generalization of RL-based reasoning. While existing work has primarily focused on investigating its generalization across tasks or modalities, this study proposes a novel cross-linguistic perspective to investigate reasoning generalization. This raises a crucial question: *Does the reasoning capability achieved from English RPT effectively transfer to other languages?* We address this by systematically evaluating English-centric LRMs on multilingual reasoning benchmarks and introducing a metric to quantify cross-lingual transferability. Our findings reveal that cross-lingual transferability varies significantly across initial model, target language, and training paradigm. Through interventional studies, we find that models with stronger initial English capabilities tend to over-rely on English-specific patterns, leading to diminished cross-lingual generalization. To address this, we conduct a thorough parallel training study. Experimental results yield three key findings: **First-Parallel Leap**, a substantial leap in performance when transitioning from monolingual to just a single parallel language, and a predictable **Parallel Scaling Law**, revealing that cross-lingual reasoning transfer follows a power-law with the number of training parallel languages. Moreover, we identify the discrepancy between actual monolingual performance and the power-law prediction as **Monolingual Generalization Gap**, indicating that English-centric LRMs fail to fully generalize across languages. Our study challenges the assumption that LRM reasoning mirrors human cognition, providing critical insights for the development of more language-agnostic LRMs.

## 1 INTRODUCTION

Recent advancements in Reinforcement Post-Training (RPT) (Jaech et al., 2024; Kimi et al., 2025; Qwen, 2025) have emerged as a transformative paradigm for advancing the capabilities of Large Reasoning Models (LRMs). Techniques like Reinforcement Learning with Verifiable Rewards (RLVR) (Lambert et al., 2024; Guo et al., 2025) have even enabled models to surpass human-level performance on complex math reasoning benchmarks such as MATH (Hendrycks et al., 2021) and AIME (Maxwell, 2024). Given these impressive gains in the mathematical domain, a central question has emerged: *Can these RL-driven reasoning abilities generalize effectively?* A growing body of work (Chu et al., 2025; Liu et al., 2025a; Hu et al., 2025a; Huan et al., 2025; Zhou et al., 2025) has investigated this by exploring generalization across tasks or modalities.

However, a crucial and largely unexplored dimension of this generalization is its cross-lingual transferability. While RL-based reasoning models have shown remarkable performance in English, it remains unclear whether these learned skills are fundamentally language-agnostic or are tied to the specific linguistic patterns of their training data. This lack of understanding regarding LRMs stands in contrast to findings from cognitive neuroscience, which have long demonstrated that human reasoning operates largely independently of language (Carruthers, 1998; Brannon, 2005; Fedorenko & Varley, 2016; Coetzee et al., 2022). In this ideal scenario, reasoning abilities should generalize across languages, as reasoning and linguistic processing are fundamentally decoupled. This provides a strong theoretical motivation for our work, which seeks to answer a critical question:

> *(Q) Does reasoning ability learned by LLMs from English training generalize to other languages, akin to human cognitive processes?*

In this work, we address this question by providing *three-stage* studies to investigate cross-lingual reasoning generalization. We start with proposing the Multilingual Transferability Index (MTI) to quantify cross-lingual transferability. We then conduct an *Observational Study*, systematically evaluating the reasoning transferability of 13 open-source English-centric LRMs spanning 11 typologically diverse languages across 4 multilingual reasoning benchmarks. This study sheds the first light on cross-lingual reasoning generalization, revealing that transferability varies significantly across the initial model, target language, and training paradigm.

Building on the initial findings of our observational study, we conducted a series of strict *Interventional Studies* to address the confounding variables present in open-source models, such as inconsistencies in training data, hyperparameters, initial models, and training paradigms. This approach allows for a precise analysis of how different training paradigms, model architectures, and model sizes influence cross-lingual generalization. Through this rigorous methodology, we found a universal principle: models with stronger initial English capabilities exhibit an over-reliance on English-specific patterns, which in turn diminishes their cross-lingual generalization.

To address this specific limitation of English-centric RPT, we conducted a comprehensive *Parallel Training Study* using parallel data from 1 to 7 languages. Through our experiments, we established three key findings: First, we identify a significant **First-Parallel Leap**, which is a substantial jump in cross-lingual generalization performance when transitioning from a monolingual to a single parallel language. Second, we uncover a predictable **Parallel Scaling Law**, which reveals that a model's multilingual reasoning performance scales in a power-law fashion with the number of parallel languages. Third, we identify a significant **Monolingual Generalization Gap**. This gap is a large discrepancy between the performance predicted by the fitted power-law function and the actual monolingual performance. The existence of this gap indicates that reasoning skills learned by English-centric LRMs are not consistent with human reasoning, as they fail to generalize completely to other languages.

In summary, we explore a new perspective on the reasoning capabilities of LLMs through the lens of cross-lingual generalizability. Our work addresses the following research questions *(RQs)* that have not been systematically examined in prior work.

- *RQ1:* To what extent do English-centric LLMs generalize their reasoning abilities across languages? (See Section 2)
- *RQ2:* What factors influence a model's cross-lingual reasoning generalization? (See Section 3)
- *RQ3:* How can we effectively improve cross-lingual reasoning generalization? (See Section 4)

## 2 OBSERVATIONAL STUDY

To address the *RQ1*, we perform an observational study by evaluating popular open-source reasoning models on diverse multilingual benchmarks. This study is designed to provide a systematic view into the cross-lingual reasoning generalization of LRMs.

### 2.1 OBSERVATIONAL SETUP

**Models** We selected a diverse set of state-of-the-art open-source LRMs, particularly those fine-tuned with Supervised Fine-Tuning (SFT) or Reinforcement Post-Training (RPT) that have demonstrated strong performance on English reasoning benchmarks. Specifically, we evaluate the Simple-Zoo (Zeng et al., 2025), s1 (Muennighoff et al., 2025), OpenThinker (Guha et al., 2025), Open-Reasoner-Zero (Hu et al., 2025b) and DeepSeek-R1-Distill (Guo et al., 2025) series models.

**Benchmarks** For evaluation, we utilized a comprehensive suite of multilingual reasoning benchmarks of challenging questions. This suite comprises multilingual version of MATH500 (Hendrycks et al., 2021), AIME2024 (Maxwell, 2024), AIME2025 (Kaggle, 2025), and GPQA-Diamond (Rein et al., 2024) from the XReasoning benchmark (Qi et al., 2025). The details of these benchmarks are described in Appendix C.1. These benchmarks are constructed from original English questions that

have been meticulously translated into ten additional languages: *Spanish (es), Russian (ru), German (de), French (fr), Bengali (bn), Swahili (sw), Thai (th), Japanese (ja), Chinese (zh), and Telugu (te)*, resulting in a total of eleven languages for evaluation.

**Experimental Setup**    Our evaluation is guided by the first principle in multilingual scenarios: *For large language models, thinking in the user's native language is as important as achieving high accuracy*. Aligning the model's reasoning language with that of the user makes its reasoning trace more readable and verifiable, which is crucial for real-world multilingual reasoning applications (Yong et al., 2025; Wang et al., 2025). Therefore, we adopted prompt hack techniques to induce models to reason in the user's language. The detailed prompt prefix provided in the Appendix F.2 follows prior work (Qi et al., 2025), which has shown that such techniques can effectively control the response language.

**Performance Metrics**    We report reasoning accuracy (**Acc**) to evaluate model performance, and the off-target rate (**Off-tag**) to measure the proportion of instances in which the LRMs fail to follow the instruction to respond in the specified language using the LangDetect library.

**Cross-lingual Transfer Metrics**    To better quantify transferability, we adopt the concept of relative gain and introduce the *Multilingual Transferability Index* (MTI), following prior work (Huan et al., 2025) that evaluated transferability across diverse tasks.

Let $S_{b,l}^{\text{trained}}$ and $S_{b,l}^{\text{base}}$ denote the accuracy score of the trained model and base model, respectively, on benchmark $b$ for language $l$. For each language $l$, we define its relative gain on benchmark $b$ as:

$$\Delta R_{b,l} = \frac{S_{b,l}^{\text{trained}} - S_{b,l}^{\text{base}}}{S_{b,l}^{\text{base}}}. \tag{1}$$

For a training language set $\mathcal{L}_{\text{train}}$ containing one or more languages (e.g., *en*, or *en & ru*), the overall relative gain is obtained by averaging over the training languages:

$$\Delta R_{b,\mathcal{L}_{\text{train}}} = \frac{1}{|\mathcal{L}_{\text{train}}|} \sum_{l \in \mathcal{L}_{\text{train}}} \Delta R_{b,l}. \tag{2}$$

The MTI for an unseen language $l_{\text{unseen}}$ (not included in the training set) on benchmark $b$ is defined as:

$$\text{MTI}_{b,l_{\text{unseen}}} = \frac{\Delta R_{b,l_{\text{unseen}}}}{\Delta R_{b,\mathcal{L}_{\text{train}}}}. \tag{3}$$

where $\Delta R_{b,l_{\text{unseen}}}$ is computed as in Eq. 1.

Finally, to obtain a single cross-lingual transferability score across all benchmarks $B$ (MATH500, AIME24/25, GPQA-Diamond), we average the per-benchmark MTI:

$$\text{MTI}_{l_{\text{unseen}}} = \frac{1}{|B|} \sum_{b \in B} \frac{\Delta R_{b,l_{\text{unseen}}}}{\Delta R_{b,\mathcal{L}_{\text{train}}}}. \tag{4}$$

A positive MTI value indicates that a model's reasoning gains have successfully transferred to the target language $l_{\text{unseen}}$, relative to its training language set $\mathcal{L}_{\text{train}}$. A value greater than 1 signifies that the reasoning gain on the target language actually exceeds that of the training languages.

## 2.2 RESULTS

Our comprehensive observational study reveals that reasoning gains acquired in English do not consistently transfer to other languages. As shown in Figure 1, the degree of transferability varies substantially across multiple dimensions, with off-target metrics results and additional details provided in Appendix E.1.

**Across Initial Models**    Our findings indicate that the choice of initial model is a critical factor influencing cross-lingual reasoning transfer. Even with the same training data, training paradigms, and hyperparameters, different initial models lead to different transfer abilities. For instance, the Qwen-2.5-7B-SimpleRL-Zoo model exhibits a slightly higher MTI than Qwen-2.5-Math-7B-SimpleRL-Zoo, despite their similar training setup. This demonstrates that **the inherent properties of the initial model influence cross-lingual transferability.**

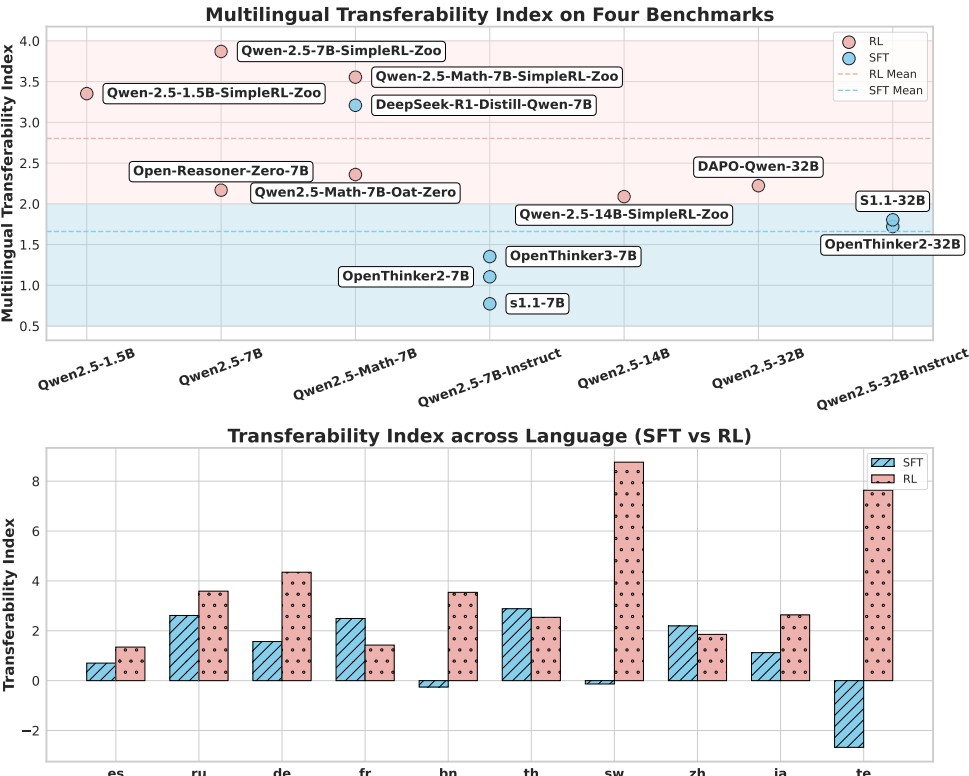

Figure 1: **Cross-lingual reasoning transferability across open-source LRMs.** The *top subfigure* shows the average Multilingual Transferability Index (MTI) of various English-centric LRMs across four benchmarks and eleven languages, with the x-axis representing the base models. The *bottom subfigure* presents the average Transferability Index (TI) performance of SFT- and RL-tuned models on individual languages on the MATH500 benchmark.

**Across Training Paradigms and Languages** In Figure 1, the top subfigure shows that RL-tuned models consistently achieve higher MTI than SFT-tuned models except DeepSeek-R1-Distill-Qwen-7B, which is fine-tuned on a massive amount of high-quality data. However, a fairer comparison requires complete control over variables, as the initial models, training parameters, and training data differ among the open-source models. The bottom subfigure illustrates that all languages except for the low-resource languages *(bn, sw, te)* exhibit positive transfer in both SFT and RL-tuned models. However, SFT and RL exhibit completely opposite effects on reasoning transfer in these low-resource languages. Specifically, SFT-tuned models exhibit negative transfer, indicating that the SFT degrades rather than enhances performance in these languages. Conversely, RL-tuned models achieve substantial positive transfer, with the TI for *sw* and *te* even exceeding that of all medium and high-resource languages. This finding suggests that RL provides a crucial solution for the low-resource language dilemma, revealing a consistent pattern: **while SFT leads to degradation in low-resource settings, RL yields substantial improvements.**

## 3 INTERVENTIONAL STUDY

While our comprehensive observational study provides an overview of existing LRMs' cross-lingual reasoning transfer capabilities, it cannot definitively isolate the underlying causes due to the varying training configurations, including datasets, training paradigms, initial model, and hyperparameters across different models.

To address *RQ2: What factors influence a model's cross-lingual reasoning generalization?*, we designed a series of interventional studies. These studies systematically control key experimental settings, enabling a more focused analysis of the isolated impacts of datasets, initial models, and training paradigms.

### 3.1 INTERVENTIONAL SETUP

**Dataset** To facilitate efficient interventional studies and inspired by prior work such as LIMO (Ye et al., 2025) and s1 (Muennighoff et al., 2025), we curated a specialized dataset. This dataset comprises 1000 samples meticulously selected from MATH training set, and all control studies are conducted using this dataset. The details of the dataset could be found in Appendix D.1.

**Training paradigm** To explore the impact of RPT on reasoning transfer, we utilize Group Rollout Policy Optimization (GRPO) (Shao et al., 2024) as our RPT algorithm. GRPO is a simplified PPO-based algorithm that significantly reduces training costs by eliminating the need for a value model. It operates by sampling $G$ rollouts $\{o_1, ..., o_G\}$ from the current policy for a given input, calculating their cumulative rewards $R = \{R_1, ..., R_G\}$, and then using these rewards to estimate advantages $\hat{A}_{i,t}$ to guide policy updates. The optimization objective for GRPO is defined as follows:

$$L_{\text{GRPO}}(\theta) = \mathbb{E}_{q \sim \mathcal{D}, \{o_i\}_{i=1}^G \sim \pi_{\theta_{\text{old}}}(\cdot|q)} \left[ \frac{1}{G} \sum_{i=1}^G \frac{1}{|o_i|} \sum_{t=1}^{|o_i|} \left( \mathcal{L}_{i,t}^{\text{clip}}(\theta) - \beta D_{\text{KL}}(\pi_\theta || \pi_{\text{ref}}) \right) \right] \quad (5)$$

where

$$\mathcal{L}_{i,t}^{\text{clip}}(\theta) = \min\left( r_{i,t}(\theta)\hat{A}_{i,t}, \ \text{clip}\left( r_{i,t}(\theta), 1 - \varepsilon, 1 + \varepsilon \right)\hat{A}_{i,t} \right)$$

$$r_{i,t}(\theta) = \frac{\pi_\theta(o_{i,t} \mid q, o_{i,<t})}{\pi_{\theta_{\text{old}}}(o_{i,t} \mid q, o_{i,<t})}, \hat{A}_{i,t} = \frac{R_i - \text{mean}(R)}{\text{std}(R)} \quad (6)$$

The clipping term with ratio $\varepsilon$ (Schulman et al., 2015) keeps the new policy close to the old one, improving training stability. In our GRPO setup, the model's policy is optimized using a composite reward function that captures reasoning accuracy $R_{\text{acc}}$, format $R_{\text{format}}$, and language consistency $R_{\text{lang}}$. Specifically, the reward $R$ for each solution is defined as a weighted sum of these three components:

$$R = \lambda_1 R_{\text{acc}} + \lambda_2 R_{\text{format}} + \lambda_3 R_{\text{lang}} \quad (7)$$

where $\lambda_{1,2,3}$ are hyperparameters controlling the relative importance of each reward component. All training hyperparameters are provided in Appendix D.3.

### 3.2 CONTROLLED SETTING AND RESULTS

For each experiment, we maintain all other hyperparameters and dataset configurations constant, only varying the specific factor.

**The Impact of Initial Model Types** To assess the influence of initial model types, we conducted controlled experiments with three distinct starting points: base model, instruction model, and math-specialized model in the Qwen2.5-7B series.

Table 1 reveals the following key findings: **(1)** The instruction model demonstrates multilingual reasoning that most aligns with real-world multilingual application scenarios, achieving the highest reasoning accuracy after training on English data (*Avg:* 23.51) and the strongest reasoning language consistency (*Off-tag*: 0.94). **(2)** When trained on English data, base and math-specific models exhibit a higher cross-lingual transferability than their instruction-tuned counterparts. Specifically, they achieved a substantially higher MTI of 1.95 and 2.12, respectively. This finding is particularly notable because these models, unlike instruction-tuned models, are not fine-tuned to be perfectly aligned with English prompts. Their superior transferability suggests that retaining more of their general pre-trained knowledge allows them to avoid over-reliance on language-specific patterns. In contrast, the strong English alignment of instruction-tuned models appears to come at the cost of cross-lingual generalization, as they become overly reliant on specific linguistic patterns.

**The Impact of Different Initial Model Families** We selected Qwen2.5-7B-Instruct and Llama3.1-8B-Instruct as our initial models to investigate the influence of the model family. Figure 2 illustrates the changes in accuracy and off-target rates from the initial models to the trained models on MATH500 benchmark. Results and analysis on more benchmarks are detailed in Appendix E.2.1. First, fine-tuning with GRPO on English data enhances LLM reasoning performance not only on the trained language but also generalizes to other languages, regardless of the model family. Interestingly, we find that the effect of cross-lingual transfer is inversely correlated with the initial model's

Table 1: **The Impact of Initial Model Type on Interventional Study.** Accuracy (%), Off-target rate (%) and MTI across different initial model types.

| Model | Average accuracy across all languages | | | | Avg | Off-tag | MTI |
|---|---|---|---|---|---|---|---|
| | *MATH500* | *AIME24* | *AIME25* | *GPQA* | | | |
| Qwen2.5-7B-Base | 26.55 | 1.23 | 0.42 | 19.79 | 12.00 | 11.41 | - |
| ↳ *GRPO on En Data* | 52.16 | 7.10 | 3.35 | 27.18 | 22.45 | 3.12 | 1.95 |
| Qwen2.5-7B-Instruct | 50.91 | 5.93 | 3.28 | 29.71 | 22.45 | 1.43 | - |
| ↳ *GRPO on En Data* | 54.24 | 7.41 | 3.92 | 28.47 | 23.51 | 0.94 | 1.23 |
| Qwen2.5-Math-7B | 29.18 | 4.11 | 1.88 | 13.82 | 12.25 | 22.59 | - |
| ↳ *GRPO on En Data* | 45.73 | 8.84 | 3.96 | 18.96 | 19.37 | 9.50 | 2.12 |

capability. Although the Llama3.1 model has a weaker initial performance on English, it demonstrates a stronger generalization ability. This finding suggests that **a model with a less specialized foundation may be better suited for broad cross-lingual transfer**. The Llama model likely possesses a more robust, less-constrained generalizable reasoning component, while the Qwen model's stronger initial performance may come from a greater reliance on language-specific patterns.

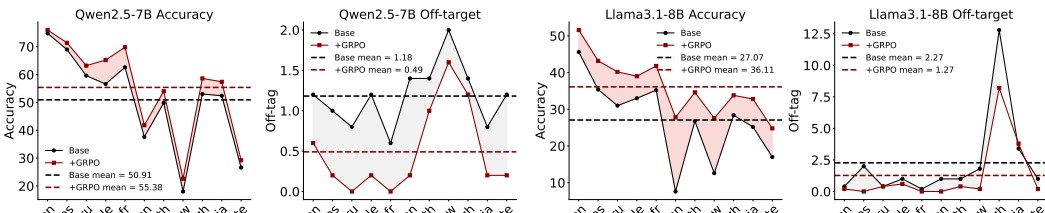

Figure 2: **The Impact of Different Initial Model Families on Interventional Study.** Multilingual reasoning performance across languages on MATH500 benchmark, comparing the influence of model family using Qwen2.5-7B-Instruct and Llama3.1-8B-Instruct as initial models. *"Base"* represents the performance of the initial model, while *"+GRPO"* denotes performance after fine-tuning with GRPO on English data. The light red area denotes the improvement in accuracy between the "Base" and "+GRPO" models, while the light gray area represents the reduction in the off-target rate between the two.

**The Impact of Model Size** To explore the multilingual performance of different model sizes, we selected the 1.5B and 7B models, which are the most common for RL training in previous research. Figure 3 shows the ΔPerformance on various multilingual benchmarks; detailed results are provided in Appendix E.2.2. On the in-domain multilingual MATH500 benchmark, the smaller 1.5B model shows substantially larger gains than the 7B model across both the training and untrained languages, indicating that **models with weaker initial capabilities achieve greater improvements on in-domain math reasoning tasks.** On the multilingual AIME24/25 benchmarks, which are used to evaluate a model's generalization to more challenging math reasoning tasks, our results show that **models with stronger initial capabilities demonstrated a more robust transfer of reasoning capabilities to these challenge benchmarks**. The multilingual GPQA-Diamond benchmark evaluates a model's reasoning capabilities in biology, physics, and chemistry. We found a clear distinction in performance between the models: 1.5B model shows significant gains on GPQA across all languages, whereas the 7B model exhibits only marginal improvements and even degradation in English.

## 4 PARALLEL TRAINING STUDY

Based on the findings from the interventional study, this section directly addresses **RQ3: *How can we effectively improve cross-lingual reasoning generalization?*** We propose a simple, yet highly efficient training strategy: "Just Go Parallel". This approach involves simultaneously training models on bilingual or more parallel problem sets in different languages. To evaluate its effectiveness, we conducted a comprehensive parallel training study analyzing how this strategy impacts cross-lingual reasoning performance.

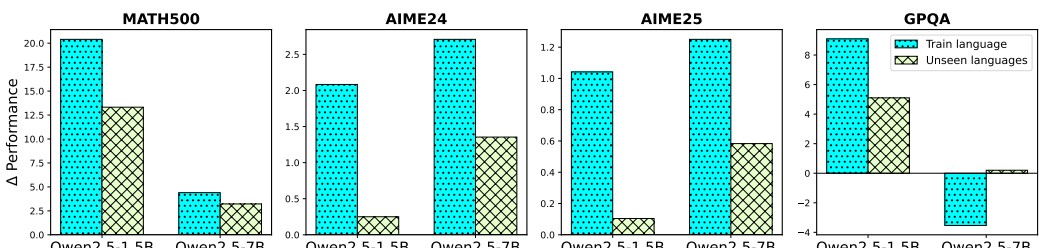

Figure 3: **The Impact of Different Model Size on Interventional Study.** Performance on various benchmarks across models of different sizes. "ΔPerformance" denotes the average difference in accuracy performance between the trained model and its initial model, averaged across both the training language and unseen languages, respectively.

## 4.1 EXPERIMENTAL SETUP AND RESULTS

We selected Qwen2.5-7B-Instruct as our initial model and fine-tuned it using the GRPO-based RPT paradigm on specialized parallel multilingual problem sets. This dataset was built from the 1,000 English samples used in our Interventional Study and extended with seven typologically diverse languages: *es, ru, de, fr, bn, th, zh*. All non-English samples were carefully aligned with their English counterparts, enabling a controlled study of parallel exposure on reasoning transfer.

To examine the effect of the number of parallel training languages on performance, we increased the number of parallel languages from one to seven, see Table 8 for details. The resulting models were evaluated on both accuracy *(Acc)* and cross-lingual transfer metrics *(MTI)* using the multilingual MATH500 benchmark across eleven languages. Figure 4 illustrates the reasoning performance and cross-lingual transferability of models trained with different numbers of parallel languages. Detailed results are present in Appendix E.3.

**The First-Parallel Leap**  We observe a striking phenomenon: the jump from monolingual to bilingual parallel languages yields a disproportionately large improvement compared to adding more parallel languages. Specifically, MTI rises from 1.16 to 2.50 (+1.34), and accuracy from 54.24 to 57.87 (+3.63). In contrast, expanding from one to seven parallel languages yields only modest gains—MTI from 2.50 to 3.63 (+1.13) and accuracy from 57.87 to 59.52 (+1.65). We term this phenomenon as the **First-Parallel Leap**, highlighting that the leap from zero to one parallel language far exceeds the cumulative gains from additional parallel languages.

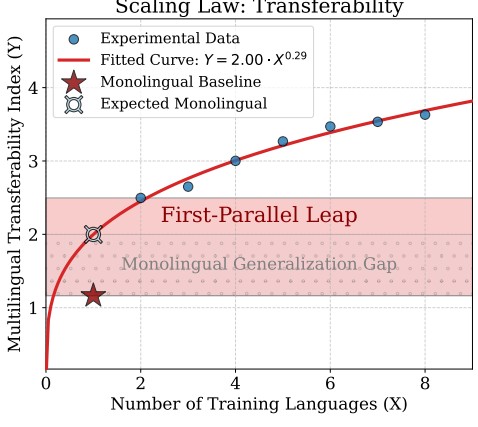
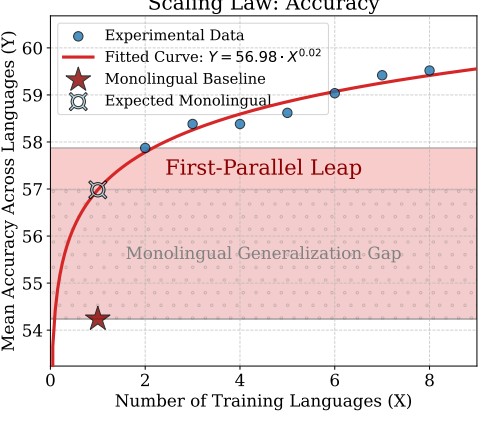

Figure 4: **The Parallel Scaling Law in Multilingual Reasoning Performance.** The x-axis *Number of Training Languages* is defined as English plus the specified number of parallel languages. *"Experimental Data"* shows the performance metrics of the model under different training numbers of parallel languages. The curves are fitted to the *Experimental Data*. *"Monolingual Baseline"* refers to fine-tuning on English data only, without parallel data. *"First-Parallel Leap"* denotes the performance difference between a model with one parallel language and the Monolingual Baseline.

**The Parallel Scaling Law**   Our observations reveal a clear scaling pattern: while the rate of improvement in both transferability and accuracy diminishes as the number of parallel languages increases from one to seven, a substantial leap in performance occurs in the initial transition from a monolingual baseline to one parallel language. This non-linear behavior, with large initial gains followed by diminishing returns, is consistent with the characteristics of power-law scaling. To model this behavior, we propose the following scaling law for cross-lingual reasoning performance, specifically for both transferability and accuracy, as a function of the number of parallel languages $X$:

$$f(X) = \alpha \cdot X^{\beta} \tag{8}$$

where $\alpha$ and $\beta$ are coefficients to be fit. Our results yield the following fitted curves for transferability and accuracy, respectively:

$$\textit{For Transferability: } f(X) = 2.00 \cdot X^{0.29} \textit{ \& For Accuracy: } f(X) = 56.98 \cdot X^{0.02} \tag{9}$$

Figure 4 presents that the fitted power-law curves demonstrate a clear and predictable scaling relationship. We term this predictable, non-linear behavior as the **Parallel Scaling Law**. Specifically, the fact that both power-law exponents are less than 1 provides mathematical proof that the model's performance gain exhibits diminishing returns as the number of parallel languages increases. The specific values of the power-law exponents ($\beta$) provide further insight. The significantly higher exponent for transferability ($\beta = 0.29$) compared to accuracy ($\beta = 0.02$) suggests that the primary benefit of parallel training is not in boosting absolute performance but in teaching the model how to transfer reasoning from English to other languages.

**Monolingual Generalization Gap**   Based on the Parallel Scaling Law, we estimate the expected performance of monolingual training (denoted as *Expected Monolingual* in Figure 4). However, when compared to the actual performance of monolingual training (denoted as *Monolingual Baseline*), a clear discrepancy emerges, which we term the **Monolingual Generalization Gap**. For instance, while the power-law fit for transferability predicts an expected monolingual MTI of approximately 2.00, the actual measured value is only 1.16. A similar gap exists for accuracy, with a predicted value of 56.98% compared to an actual value of 54.24%. This gap reveals a crucial insight: the reasoning abilities acquired by English-centric models through monolingual training do not adhere to the same scaling behavior observed in multilingual training. This indicates that these English-centric models, despite their impressive capabilities, are likely relying on language-specific patterns rather than a universal, language-agnostic reasoning component.

### 4.2 ANALYSIS AND DISCUSSION

**Parallel vs. Unparallel**   The use of parallel data is a critical component of our proposed training strategy. Unlike unparallel data, which simply exposes the model to a wider variety of languages, parallel data provides an explicit signal for semantic equivalence across languages. This forces the model to learn a unified, language-agnostic representation for reasoning. Figure 5 shows the performance differences between using parallel and non-parallel data, highlighting the critical importance of training with parallel data.

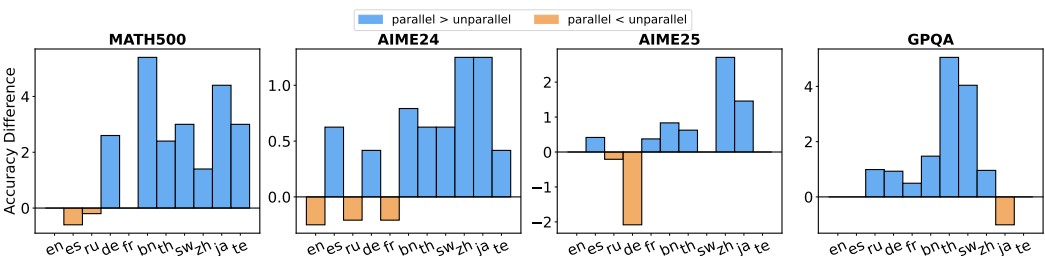

Figure 5: Accuracy difference comparison across parallel and unparallel data training.

**Is the selected language important for the parallel training?**   Figure 6 shows that the choice of parallel language results in only minor variations in MTI and off-target metrics. Among the parallel languages, training with Russian achieves the highest MTI of 2.84 (higher than Bengali at 2.73,

German at 2.56, and Chinese at 2.50) and also attains the lowest off-target rate. Overall, adding one parallel language consistently enhances both cross-lingual transferability and multilingual reasoning performance. More detailed analyses are presented in Appendix E.3.4.

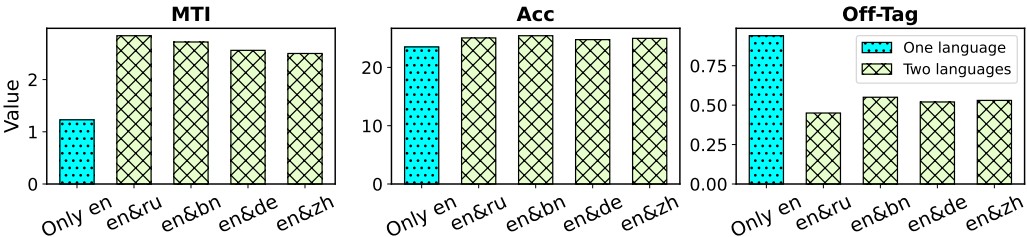

Figure 6: Multilingual reasoning performance across different parallel languages. "Only en" denotes only fine-tuned on English data. "en&LANGUAGE" indicates the model was fine-tuned on English and a parallel language, with LANGUAGE representing *ru, bn, de, zh*, respectively.

## 5 RELATED WORK

**Large Reasoning Models**   Chain of Thought (CoT) (Wei et al., 2022) has unlocked the reasoning capabilities of large language models (LLMs), enabling step-wise thought and improving reasoning performance. OpenAI's O1 (Jaech et al., 2024) marked a paradigm shift by using reinforcement learning (RL) for test-time scaling, simulating human-like reflective reasoning. Building on this, DeepSeek R1 (Guo et al., 2025) employed GRPO (Shao et al., 2024) with rule-based rewards, fostering long CoT sequences and self-reflection. These advances have sparked a wave of open-source efforts to replicate or extend R1's methods and refine RL algorithms (Liu et al., 2025b; Yu et al., 2025; Sun et al., 2025).

**Reasoning Generalization**   Reasoning generalization in RL-based LLMs has attracted growing interest, particularly in transferring mathematical reasoning to other tasks or modalities. Hu et al. (2025a) shows that RL improves structured reasoning but transfers poorly to unstructured tasks. Huan et al. (2025); Chu et al. (2025) find that RL encourages broader transfer, whereas SFT often leads to domain-specific overfitting. X-REASONER (Liu et al., 2025a) demonstrates that rule-based RL can generalize reasoning across domains and modalities. While prior works explore reasoning generalization across domains and modalities, our work proposes a new cross-linguistic perspective to investigate reasoning generalization.

**Cross-Lingual Transferability**   Improving the performance of English-centric LLMs in other languages has become a major research focus. Prior work has explored zero-shot or minimal fine-tuning to realize cross-lingual transfer (Li et al., 2024; Chirkova & Nikoulina, 2024), showing that English reward models (Wu et al., 2024; Hong et al., 2024) and preference alignment (Yang et al., 2024b; 2025) can generalize across languages. In the era of reasoning, Bandarkar et al. (2024) transfers math skills to other languages by swapping a few layers between a math-specific and multilingual model. Yong et al. (2025) demonstrates that cross-lingual test-time scaling improves multilingual reasoning. Distinct from these studies, our work adopts a cross-lingual perspective to systematically analyze the reasoning generalization of RL-based models.

## 6 CONCLUSION

This work presents a systematic study of cross-lingual reasoning generalization in English-centric LRMs. Through observational and interventional studies, we reveal that stronger English-centric models often overfit to language-specific patterns, limiting cross-lingual transfer. In our parallel training study, we uncover three key phenomena that characterize cross-lingual reasoning: *First-Parallel Leap*, *Parallel Scaling Law*, and *Monolingual Generalization Gap*, providing a principled framework for enhancing cross-lingual reasoning generalization. These results highlight both the limitations of current LRMs and shed light on building more language-agnostic LRMs.

## REPRODUCIBILITY STATEMENT

Codes and model weights will be released after review to facilitate future research. For evaluation, we follow prior works and report averaged results over 16 sampled generations per question on data-scarce benchmarks. All evaluations are conducted with temperature set to 0.6 and top-p to 0.95, with the random seed fixed to ensure deterministic outputs across runs. Note that minor variations in inference results may still occur due to differences in hardware or the version of the inference framework.

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

APPENDIX

## A   LIMITATIONS AND FUTURE WORK

**Generalizability and Domain Expansion**   A primary limitation of our work is its focus on the mathematical reasoning domain. While our findings on the *Parallel Scaling Law* and the *Just Go Parallel* strategy reveal a valuable phenomenon, their generalizability to other domains, such as coding, agent planning, remains to be verified. Future work could explore the extent to which our *Parallel Scaling Law* holds true across different domains. A key direction is to develop more sophisticated parallel training strategies that can better overcome the diminishing returns shown in the scaling law. Furthermore, investigating how to apply our findings to the challenge of low-resource languages, a key dilemma identified in our observational study, would be a critical next step.

**Interpretability**   While our interventional and parallel studies provide strong evidence for the correlations between initial model capabilities, training strategies, and cross-lingual transfer, the underlying causal mechanisms are more hypothetical. For instance, the precise reason for RL's advantage in low-resource languages or the specific linguistic patterns that hinder cross-lingual transfer remain open questions. Future work could focus on mechanistic interpretability to dissect the internal representations and analyze specific failure modes, providing a deeper understanding of how reasoning and language are coupled in these models.

## B   THE USAGE OF LARGE LANGUAGE MODEL

We declare that the LLM was only used to refine the fluency of certain sentences during the writing of this paper. Every sentence polished with the LLM was carefully reviewed and approved by the authors. The LLM was not used for any other part of this research.

## C   EVALUATION DETAILS AND SETUP

### C.1   MULTILINGUAL REASONING BENCHMARKS

We use the multilingual version of these four reasoning benchmarks provided in Qi et al. (2025), which use GPT-4o-MINI (Jaech et al., 2024) to translate all questions into the other ten languages *Spanish (es), Russian (ru), German (de), French (fr), Bengali (bn), Swahili (sw), Thai (th), Japanese (ja), Chinese (zh), and Telugu (te)*, resulting in a total of eleven languages for evaluation.

**MATH500**   The MATH500 (Hendrycks et al., 2021) benchmark assesses the mathematical reasoning and problem-solving abilities of language models, addressing the need for more challenging evaluations as their general capabilities advance. It consists of 500 problems across five core mathematical domains: algebra, combinatorics, geometry, number theory, and precalculus. Each problem is designed to test multi-step reasoning and complex problem-solving skills, going beyond simple calculations or factual recall.

**AIME24&25**   The AIME24 (Maxwell, 2024) and AIME25 (Kaggle, 2025) datasets contain problems from the American Invitational Mathematics Examination (AIME) for 2024 and 2025, respectively. AIME is a prestigious high school mathematics competition renowned for its challenging problems, consisting of 30 questions.

**GPQA-Diamond**   GPQA-Diamond (Rein et al., 2024) consists of 198 multiple-choice questions across biology, chemistry, and physics, with difficulty levels ranging from challenging undergraduate to postgraduate. It is the highest quality subset, which includes only questions where both experts answer correctly and the majority of non-experts answer incorrectly.

### C.2   AN OVERVIEW OF OPEN-SOURCE LRMS

Table 2 provides an overview of the various open-source LLMs evaluated in our observational study. These models, which include the DeepSeek-R1-Distill-Qwen-7B (Guo et al., 2025), OpenThinker

series (Guha et al., 2025), Simple-RL-Zoo series (Zeng et al., 2025), s1 series (Muennighoff et al., 2025), and DAPO-Qwen-32B (Yu et al., 2025), range in size from 1.5B to 32B.

Table 2: **The Overview of the Open-source LLMs Used in Observational Study**, including their initial model, parameter size, and training paradigm.

| Model | Initial Model | Size | Training Paradigm |
|---|---|---|---|
| DeepSeek-R1-Distill-Qwen-7B | Qwen2.5-Math-7B-Base | 7B | SFT |
| Open-Reasoner-Zero-7B | Qwen2.5-7B-Base | 7B | RL |
| OpenThinker2-7B | Qwen2.5-7B-Instruct | 7B | SFT |
| OpenThinker3-7B | Qwen2.5-7B-Instruct | 7B | SFT |
| Qwen-2.5-1.5B-SimpleRL-Zoo | Qwen2.5-1.5B-Base | 1.5B | RL |
| Qwen-2.5-7B-SimpleRL-Zoo | Qwen2.5-7B-Base | 7B | RL |
| Qwen-2.5-14B-SimpleRL-Zoo | Qwen2.5-14B-Base | 14B | RL |
| Qwen-2.5-Math-7B-SimpleRL-Zoo | Qwen2.5-Math-7B-Base | 7B | RL |
| Qwen2.5-Math-7B-Oat-Zero | Qwen2.5-Math-7B-Base | 7B | RL |
| s1.1-7B | Qwen2.5-7B-Instruct | 7B | SFT |
| DAPO-Qwen-32B | Qwen2.5-32B-Base | 32B | RL |
| OpenThinker2-32B | Qwen2.5-32B-Instruct | 32B | SFT |
| s1.1-32B | Qwen2.5-32B-Instruct | 32B | SFT |

# D    IMPLEMENTATION DETAILS

## D.1    TRAINING DATASET

**The Distribution of Parallel Questions**    Figure 7a shows the type and level distributions of the 1,000 English training questions sampled from the MATH dataset (Hendrycks et al., 2021). The type distribution is relatively balanced, and the number of questions increases steadily from Level 1 to Level 5.

**The Distribution of Unparallel Questions**    Moreover, Figure 7b presents the type and level distributions of 1,000 Russian questions used for an unparalleled training analysis experiment, which form a separate, non-overlapping set from the 1000 English questions. The distributions of both type and level closely match those of the English training questions. This indicates that, in the analysis comparing parallel and unparallel training, the performance drop observed in unparallel training is not due to distributional differences between the unparallel and English datasets.

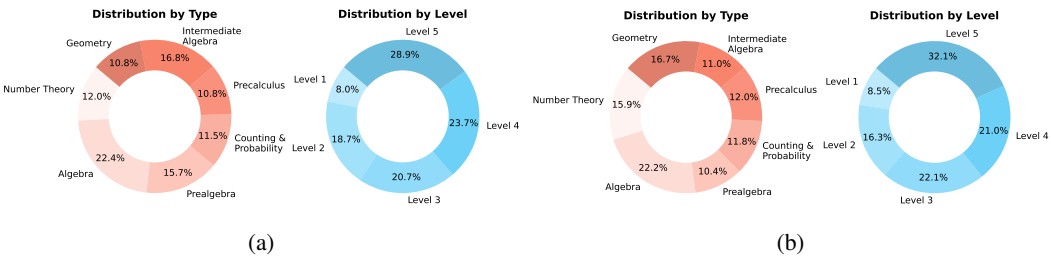

Figure 7: Distribution of question difficulty. (a) The 1,000 English questions were utilized in the interventional study and for parallel training. (b) The 1,000 Russian questions for unparalleled training, comprising a separate and non-overlapping set from the questions in (a).

## D.2    EXPERIMENTS ENVIRONMENTS

All training and inference experiments were conducted on Ubuntu 22.04 equipped with $8 \times$ NVIDIA A800 GPUs. For RL training, we RL-tune all models using VeRL v0.2 (Sheng et al., 2024) with

customized rewards. For Inference, we performed with `vLLM 0.8.5` (Kwon et al., 2023). For Evaluation, we used Qwen's Math codebase (Yang et al., 2024a) for evaluation, following the prior work (Zeng et al., 2025; Liu et al., 2025b).

### D.3 HYPERPARAMETERS

**RL Training**  The maximum generation length was set to 4096 tokens, and the maximum prompt length to 1024 tokens, such that their sum matches the model's maximum context length. The learning rate was fixed at $1 \times 10^{-6}$. Training was performed with a batch size of 128 questions. For each question, $G = 16$ rollouts were sampled, using a sampling temperature of 1.0. $\lambda_1 = 0.8$, $\lambda_2 = 0.1$, and $\lambda_3 = 0.1$.

**Inference**  In the evaluation setup, we used a temperature of 0.6, a top-$p$ value of 0.95, and a maximum generation length of 8912 tokens for all models in the 1.5B–14B series. For 32B models, we used the same temperature (0.6) and top-$p$ value (0.95), but set the maximum generation length to 32,768 tokens, except for DAPO-Qwen-32B, which followed the official recommended settings: a temperature of 1.0, a top-$p$ value of 0.7, and a maximum generation length of 20,480 tokens. For AIME2024 and AIME2025, we report accuracy by averaging over 16 sampled generations per question, while for MATH500 and GPQA, accuracy is computed using a single sampled generation per question.

## E  DETAILED RESULTS AND ANALYSIS

### E.1  OBSERVATIONAL STUDY

#### E.1.1  HOW TO SELECT A TEMPLATE FOR BASE MODEL?

To accurately measure the reasoning capabilities of our base models for the transfer efficiency calculation, we evaluated them across different template settings. Specifically, we tested the *Qwen2.5-7B-Base* and *Qwen2.5-Math-7B-Base* models using `Qwen-Math Template`, `Qwen-Instruct Template`, and `No Template` setting.

---

**Qwen-Instruct Template:**

```
<|im_start|>system\n
You are Qwen, created by Alibaba Cloud. You are a helpful assistant.
<|im_end|>\n
<|im_start|>user\n{instruction}<|im_end|>\n
<|im_start|>assistant\n
```

**Qwen-Math Template:**

```
<|im_start|>system\n
Please reason step by step, and put your final answer within \boxed{}.
<|im_end|>\n
<|im_start|>user\n{instruction}<|im_end|>\n
<|im_start|>assistant\n
```

**No Template:**

```
{instruction}
```

---

As shown in Table 3, the `Qwen-Instruct Template` consistently yielded better reasoning accuracy and improved reasoning language consistency for both base models on the multilingual MATH500 benchmark. This result guided our decision to use the `Qwen-Instruct Template` as the default for evaluating all math-based and general-based models.

### E.1.2  THE PERFORMANCE OF INITIAL MODELS

To address the lack of detailed analysis on the influence of initial model properties on cross-lingual transfer and multilingual reasoning, we conducted a comprehensive evaluation of various base mod-

Table 3: **The Performance of Base Models with Different Template Settings.** Accuracy (%) and Off-target rate (%) across languages for different template settings on multilingual MATH500 benchmark.

| Settings | Accuracy per language | | | | | | | | | | | Average | |
| | en | es | ru | de | fr | bn | th | sw | zh | ja | te | Acc | Off-tag |
|---|---|---|---|---|---|---|---|---|---|---|---|---|---|
| *Qwen2.5-7B-Base* | | | | | | | | | | | | | |
| Qwen-Math Template | 49.2 | 31.8 | 25.2 | 28.2 | 30.0 | 5.8 | 23.0 | 2.4 | 27.0 | 15.2 | 1.4 | 21.7 | 16.6 |
| Qwen-Instruct Template | 50.6 | 38.0 | 30.0 | 33.2 | 38.4 | 10.4 | 26.8 | 2.4 | 30.0 | 27.8 | 4.4 | 26.5 | 15.7 |
| No Template | 44.4 | 38.2 | 28.2 | 28.4 | 35.6 | 6.0 | 17.0 | 0.2 | 29.2 | 19.8 | 1.2 | 22.6 | 18.5 |
| *Qwen2.5-Math-7B-Base* | | | | | | | | | | | | | |
| Qwen-Math Template | 43.4 | 36.6 | 2.8 | 21.8 | 21.4 | 26.2 | 15.0 | 2.2 | 36.6 | 9.4 | 1.2 | 19.7 | 30.5 |
| Qwen-Instruct Template | 56.6 | 46.4 | 11.2 | 33.4 | 36.8 | 31.4 | 28.2 | 4.2 | 44.4 | 25.2 | 3.2 | 29.2 | 18.0 |
| No Template | 37.8 | 33.0 | 4.2 | 29.8 | 37.4 | 29.0 | 12.4 | 0.0 | 36.2 | 17.2 | 3.0 | 21.8 | 31.5 |

els. Table 4 presents the detailed results of this analysis, including the accuracy and off-target rate across languages.

Our evaluation reveals several key insights from the Qwen2.5-7B series. From a linguistic perspective, the Instruct model exhibits the lowest off-target rate, followed by the General Base model and the Math Base model. However, when evaluated on multilingual reasoning accuracy, the order is reversed: the Instruct model significantly outperforms the Math Base model, which in turn performs better than the General Base model.

Furthermore, a clear scaling trend is observed within the Qwen2.5-Base models. As model size increases from 1.5B to 32B, the off-target rate decreases while multilingual reasoning accuracy steadily improves.

A particularly striking finding is that the Qwen2.5-7B-Instruct model achieves greater multilingual reasoning accuracy than the much larger Qwen2.5-32B model. This suggests that the instruction-following capability is a critical factor for activating a model's multilingual reasoning abilities.

This result challenges the popular wisdom that math-specific models are more amenable to RL training, especially when viewed through the lens of cross-lingual reasoning transfer. We posit that the superior multilingual capabilities of general instruction models make them a more suitable initial model for RL training compared to both general base and math base models. These results highlight instruction-tuned models as the most advantageous starting point for enhancing multilingual reasoning through RL.

### E.1.3 THE PERFORMANCE OF OPEN-SOURCE MODELS

Tables 5 and 6, in conjunction with Figure 8, provide a detailed analysis of open-source model performance. These results illustrate the Multilingual Transferability Index (MTI), accuracy, and off-target rates of open-source models, and highlight the distinct performance differences between RL-tuned and SFT-tuned models across languages.

Figure 8a further shows that all 7B SFT-tuned models exhibit performance degradation on *bn*, *sw*, and *te*, with the sole exception of DeepSeek-R1-Distill-Qwen-7B. Notably, this model was fine-tuned on a massive amount of high-quality data generated by the DeepSeek-R1 model. Scaling the model size up to 32B provides modest performance gains across most languages, suggesting that larger models can partially mitigate the negative effects of SFT. However, the degradation in low-resource languages remains unresolved, as evidenced by the performance drop of OpenThinker-32B on *sw*. Figure 8b shows that all RL-tuned models improve performance across all languages, with particularly large gains in *bn*, *sw*, and *te*. The heatmaps in Figure 8a and Figure 8b clearly illustrate the performance gap between SFT-tuned and RL-tuned models on low-resource languages, revealing a consistent pattern: **while SFT leads to degradation in low-resource settings, RL yields substantial improvements.**

Table 4: **The Performance of Initial Models.** Accuracy (%) and Off-target rate (%) across languages for different Initial models.

| Settings | Accuracy per language | | | | | | | | | | | Average | |
|---|---|---|---|---|---|---|---|---|---|---|---|---|---|
| | en | es | ru | de | fr | bn | th | sw | zh | ja | te | Acc | Off-tag |
| *Multilingual MATH500* | | | | | | | | | | | | | |
| Qwen2.5-1.5B | 19.60 | 10.60 | 7.80 | 1.00 | 9.80 | 1.00 | 3.00 | 0.00 | 8.60 | 2.20 | 0.00 | 5.78 | 21.02 |
| Qwen2.5-Math-7B | 56.60 | 46.40 | 11.20 | 33.40 | 36.80 | 31.40 | 28.20 | 4.20 | 44.40 | 25.20 | 3.20 | 29.18 | 17.96 |
| Qwen2.5-7B-Instruct | 74.80 | 69.00 | 59.60 | 56.60 | 62.60 | 37.60 | 49.80 | 18.00 | 53.00 | 52.40 | 26.60 | 50.91 | 0.18 |
| Qwen2.5-7B | 50.60 | 38.00 | 30.00 | 33.20 | 38.40 | 10.40 | 26.80 | 2.40 | 30.00 | 27.80 | 4.40 | 26.55 | 15.69 |
| Qwen2.5-14B | 42.20 | 40.40 | 36.00 | 29.60 | 36.20 | 22.60 | 26.60 | 5.60 | 18.80 | 25.20 | 5.60 | 26.25 | 15.55 |
| Qwen2.5-32B | 54.00 | 50.80 | 42.00 | 37.80 | 46.80 | 24.60 | 33.00 | 13.60 | 28.00 | 42.60 | 11.60 | 34.98 | 5.56 |
| Qwen2.5-32B-Instruct | 78.60 | 73.60 | 68.00 | 68.40 | 69.40 | 53.20 | 60.60 | 37.40 | 61.40 | 65.00 | 43.40 | 61.73 | 0.13 |
| *Multilingual AIME24* | | | | | | | | | | | | | |
| Qwen2.5-1.5B | 0.21 | 0.42 | 0.00 | 0.00 | 0.00 | 0.00 | 0.21 | 0.00 | 0.63 | 0.00 | 0.00 | 0.13 | 19.26 |
| Qwen2.5-Math-7B | 13.75 | 6.46 | 1.67 | 3.33 | 4.79 | 2.71 | 3.33 | 0.00 | 6.88 | 1.67 | 0.63 | 4.11 | 21.76 |
| Qwen2.5-7B-Instruct | 10.42 | 8.96 | 8.13 | 8.75 | 8.54 | 2.92 | 4.58 | 1.04 | 5.63 | 4.38 | 1.88 | 5.93 | 0.34 |
| Qwen2.5-7B | 2.29 | 1.67 | 2.08 | 2.71 | 2.08 | 0.21 | 0.63 | 0.00 | 1.25 | 0.63 | 0.00 | 1.23 | 9.89 |
| Qwen2.5-14B | 2.50 | 2.50 | 2.29 | 2.50 | 2.71 | 0.63 | 0.21 | 0.00 | 1.04 | 0.83 | 0.21 | 1.40 | 16.02 |
| Qwen2.5-32B | 2.71 | 3.13 | 2.08 | 3.33 | 2.71 | 0.21 | 1.04 | 0.00 | 1.25 | 2.08 | 0.00 | 1.69 | 4.07 |
| Qwen2.5-32B-Instruct | 15.63 | 12.71 | 11.25 | 11.67 | 12.08 | 5.42 | 7.50 | 2.92 | 7.29 | 10.63 | 2.50 | 9.05 | 0.51 |
| *Multilingual AIME25* | | | | | | | | | | | | | |
| Qwen2.5-1.5B | 0.00 | 0.00 | 0.00 | 0.00 | 0.00 | 0.00 | 0.00 | 0.00 | 0.21 | 0.00 | 0.00 | 0.02 | 61.14 |
| Qwen2.5-Math-7B | 6.04 | 3.13 | 0.83 | 1.46 | 2.29 | 0.42 | 0.83 | 0.00 | 4.79 | 0.42 | 0.42 | 1.88 | 23.47 |
| Qwen2.5-7B-Instruct | 7.08 | 5.63 | 5.21 | 3.96 | 4.79 | 0.83 | 1.67 | 0.00 | 3.75 | 2.71 | 0.42 | 3.28 | 0.55 |
| Qwen2.5-7B | 0.83 | 0.83 | 0.21 | 1.25 | 0.63 | 0.21 | 0.00 | 0.00 | 0.42 | 0.21 | 0.00 | 0.42 | 10.38 |
| Qwen2.5-14B | 1.25 | 2.08 | 2.50 | 1.67 | 1.46 | 0.00 | 0.21 | 0.00 | 0.42 | 1.46 | 0.21 | 1.02 | 16.99 |
| Qwen2.5-32B | 1.04 | 1.04 | 1.46 | 0.21 | 0.83 | 0.21 | 0.00 | 0.00 | 1.04 | 1.04 | 0.21 | 0.64 | 4.02 |
| Qwen2.5-32B-Instruct | 11.25 | 7.29 | 7.29 | 6.04 | 6.67 | 1.25 | 2.71 | 0.00 | 5.42 | 2.71 | 0.42 | 4.64 | 0.30 |
| *Multilingual GPQA-Diamond* | | | | | | | | | | | | | |
| Qwen2.5-1.5B | 15.66 | 14.65 | 15.15 | 16.67 | 11.62 | 7.58 | 15.15 | 13.64 | 15.15 | 6.06 | 15.15 | 13.31 | 20.02 |
| Qwen2.5-Math-7B | 16.16 | 13.64 | 3.54 | 17.17 | 15.66 | 17.68 | 17.68 | 11.62 | 22.22 | 0.51 | 16.16 | 13.82 | 27.18 |
| Qwen2.5-7B-Instruct | 36.36 | 32.83 | 23.74 | 36.36 | 33.84 | 26.77 | 29.80 | 24.75 | 30.81 | 29.80 | 21.72 | 29.71 | 0.64 |
| Qwen2.5-7B | 28.79 | 24.24 | 20.71 | 22.22 | 18.18 | 11.11 | 21.72 | 17.68 | 23.23 | 17.17 | 12.63 | 19.79 | 9.69 |
| Qwen2.5-14B | 26.26 | 15.15 | 20.71 | 21.21 | 27.78 | 20.20 | 24.24 | 22.22 | 12.12 | 10.61 | 16.16 | 19.70 | 20.98 |
| Qwen2.5-32B | 28.28 | 30.30 | 28.28 | 33.33 | 23.74 | 15.66 | 24.75 | 17.68 | 27.27 | 27.78 | 17.17 | 24.93 | 9.00 |
| Qwen2.5-32B-Instruct | 45.45 | 41.92 | 38.89 | 41.41 | 44.44 | 29.80 | 38.38 | 32.83 | 36.36 | 38.38 | 27.27 | 37.74 | 0.28 |

(a) SFT-tuned Models

(b) RL-tuned Models

Figure 8: **The Performance of Various Open-source Models. Part 2:** The transferability difference between SFT-tuned and RL-tuned models across languages. Note that "*None*" values indicate that *Qwen-2.5-1.5B* achieved zero accuracy in most languages on AIME24 and AIME25, making relative gain undefined.

## E.2 INTERVENTIONAL STUDY

### E.2.1 THE IMPACT OF DIFFERENT MODEL FAMILIES

Figure 9 compares the influence of model family on cross-lingual reasoning by using Qwen2.5-7B-Instruct and Llama3.1-8B-Instruct as initial models. We find that reinforcement learning (RL)

Table 5: **The Performance of Various Open-source Models. Part 1:** Multilingual Transferability Index (MTI) of various models across benchmarks. The columns ID, OOD, and Avg refer to the MTI on in-domain (MATH500), out-of-domain (AIME24, AIME25, GPQA-Diamond), and all tasks, respectively. Note that "*None*" values indicate that *Qwen-2.5-1.5B* achieved zero accuracy in most languages on AIME24 and AIME25, making relative gain undefined.

| Models | Multilingual Reasoning Benchmarks | | | | MTI | | |
|---|---|---|---|---|---|---|---|
| | *MATH500* | *AIME24* | *AIME25* | *GPQA-D* | *ID* | *OOD* | *Avg* |
| DeepSeek-R1-Distill-Qwen-7B | 3.493 | 2.312 | 2.864 | 4.168 | 3.493 | 3.115 | 3.209 |
| Open-Reasoner-Zero-7B | 3.195 | 2.677 | 1.479 | 1.320 | 3.195 | 1.825 | 2.168 |
| OpenThinker2-7B | 0.093 | 0.876 | 1.843 | 1.604 | 0.093 | 1.441 | 1.104 |
| OpenThinker3-7B | 0.157 | 1.502 | 2.434 | 1.318 | 0.157 | 1.752 | 1.353 |
| Qwen-2.5-1.5B-SimpleRL-Zoo | 5.322 | *None* | *None* | 1.383 | 5.322 | 1.383 | 3.353 |
| Qwen-2.5-7B-SimpleRL-Zoo | 4.543 | 3.189 | 1.217 | 6.531 | 4.543 | 3.646 | 3.870 |
| Qwen-2.5-14B-SimpleRL-Zoo | 2.381 | 3.360 | 0.959 | 1.655 | 2.381 | 1.991 | 2.089 |
| Qwen-2.5-Math-7B-SimpleRL-Zoo | 3.920 | 2.884 | 3.079 | 4.335 | 3.920 | 3.433 | 3.555 |
| Qwen2.5-Math-7B-Oat-Zero | 2.807 | 1.324 | 3.158 | 2.149 | 2.807 | 2.210 | 2.359 |
| s1.1-7B | 0.310 | 0.920 | 1.192 | 0.671 | 0.310 | 0.928 | 0.773 |
| DAPO-Qwen-32B | 3.634 | 2.337 | 2.066 | 0.854 | 3.634 | 1.752 | 2.223 |
| OpenThinker2-32B | 0.936 | 1.513 | 4.235 | 0.201 | 0.936 | 1.983 | 1.721 |
| S1.1-32B | 1.382 | 1.583 | 3.429 | 0.821 | 1.382 | 1.944 | 1.804 |

consistently improves reasoning performance across all languages, regardless of the initial model family.

However, a notable difference is that Llama3.1 exhibits a substantially greater performance gain on various benchmarks compared to Qwen2.5. This result suggests a counter-intuitive principle: **models with weaker initial English capabilities may possess greater potential for cross-lingual generalization**. We posit that while stronger English-capable models, such as Qwen2.5, excel at English reasoning, they may become too entrenched in English-specific reasoning patterns, thereby limiting their ability to transfer these skills to other languages.

### E.2.2 THE IMPACT OF MODEL SIZE

Table 7 presents the detailed results of our controlled study on model scaling, comparing the performance of *Qwen2.5-1.5B-Instruct* and *Qwen2.5-7B-Instruct* as initial models.

We found a clear distinction in transferability based on model size. The smaller 1.5B model exhibits larger relative gains on its in-domain training task (MATH500) and out-of-domain tasks (GPQA-Diamond), likely due to its weaker initial capabilities. In contrast, the larger 7B model shows smaller training gains in MATH500 and GPQA-Diamond but demonstrates superior transfer to more challenging tasks such as AIME24 and AIME25.

This observation suggests a key trade-off: **models with stronger initial English performance have less potential for large relative gains in cross-lingual generalization, whereas smaller, weaker models possess a greater capacity for significant improvement across languages.**

### E.3 PARALLEL SCALING LAW

### E.3.1 THE LANGUAGE SETTINGS IN PARALLEL SCALING LAW

Table 8 outlines the language settings for our experiment to validate the parallel scaling law, systematically increasing the number of parallel languages from 1 to 7.

Table 6: **The Performance of Various Open-source Models. Part 3:** Accuracy (%) and Off-target rate (%) across languages for various open-source models.

| Settings | Accuracy per language | | | | | | | | | | | Average | |
|---|---|---|---|---|---|---|---|---|---|---|---|---|---|
| | en | es | ru | de | fr | bn | th | sw | zh | ja | te | Acc | Off-tag |
| *Multilingual MATH500* | | | | | | | | | | | | | |
| DeepSeek-R1-Distill-Qwen-7B | 86.20 | 76.20 | 67.00 | 66.40 | 69.80 | 40.20 | 53.80 | 18.40 | 70.00 | 53.20 | 17.60 | 56.25 | 2.69 |
| Open-Reasoner-Zero-7B | 81.60 | 76.00 | 70.40 | 69.80 | 71.20 | 45.20 | 63.20 | 15.00 | 62.80 | 61.80 | 17.60 | 57.69 | 5.20 |
| OpenThinker2-7B | 86.00 | 74.80 | 64.80 | 63.00 | 73.00 | 34.40 | 58.80 | 12.60 | 65.80 | 70.00 | 8.40 | 55.60 | 36.13 |
| OpenThinker3-7B | 85.80 | 81.80 | 75.00 | 69.20 | 76.40 | 17.00 | 48.80 | 17.40 | 69.60 | 65.00 | 10.40 | 56.04 | 60.75 |
| Qwen-2.5-1.5B-SimpleRL-Zoo | 57.60 | 41.40 | 36.80 | 38.20 | 42.40 | 11.80 | 28.80 | 14.60 | 33.60 | 31.00 | 7.40 | 31.24 | 21.51 |
| Qwen-2.5-7B-SimpleRL-Zoo | 77.60 | 72.00 | 65.00 | 64.20 | 68.00 | 42.20 | 60.40 | 24.20 | 62.00 | 59.80 | 25.80 | 56.47 | 4.31 |
| Qwen-2.5-14B-SimpleRL-Zoo | 82.40 | 74.40 | 71.00 | 69.40 | 73.60 | 54.80 | 69.00 | 33.20 | 65.60 | 68.80 | 41.00 | 63.93 | 0.47 |
| Qwen-2.5-Math-7B-SimpleRL-Zoo | 80.40 | 72.60 | 66.00 | 68.60 | 70.60 | 45.80 | 57.40 | 15.00 | 61.80 | 54.40 | 14.20 | 55.16 | 8.33 |
| Qwen2.5-Math-7B-Oat-Zero | 79.80 | 72.40 | 32.80 | 55.60 | 50.40 | 47.60 | 49.40 | 16.80 | 58.00 | 43.40 | 11.80 | 47.09 | 15.11 |
| s1.1-7B | 75.80 | 68.20 | 60.80 | 59.00 | 69.60 | 37.00 | 57.40 | 16.40 | 57.20 | 51.60 | 20.40 | 52.13 | 12.76 |
| DAPO-Qwen-32B | 68.80 | 65.00 | 58.80 | 60.20 | 63.00 | 52.80 | 58.40 | 44.80 | 54.20 | 56.80 | 44.80 | 57.05 | 11.85 |
| OpenThinker2-32B | 96.00 | 88.60 | 85.20 | 84.20 | 85.00 | 73.00 | 80.60 | 35.40 | 77.40 | 75.60 | 47.20 | 75.29 | 13.02 |
| S1.1-32B | 95.40 | 91.20 | 85.00 | 83.60 | 88.00 | 73.00 | 81.20 | 57.00 | 77.40 | 80.80 | 53.60 | 78.75 | 27.40 |
| *Multilingual AIME24* | | | | | | | | | | | | | |
| DeepSeek-R1-Distill-Qwen-7B | 40.63 | 27.71 | 26.25 | 23.13 | 27.50 | 6.46 | 9.79 | 2.50 | 30.42 | 9.79 | 0.83 | 18.64 | 7.77 |
| Open-Reasoner-Zero-7B | 16.25 | 18.13 | 17.29 | 15.21 | 17.71 | 9.58 | 14.79 | 1.67 | 14.79 | 14.79 | 1.25 | 12.86 | 10.78 |
| OpenThinker2-7B | 37.08 | 18.33 | 17.08 | 13.75 | 20.83 | 11.04 | 20.21 | 2.50 | 25.63 | 27.29 | 5.42 | 18.11 | 39.72 |
| OpenThinker3-7B | 26.25 | 32.08 | 23.54 | 26.46 | 29.38 | 5.63 | 16.25 | 3.54 | 26.46 | 19.38 | 3.54 | 19.32 | 63.28 |
| Qwen-2.5-1.5B-SimpleRL-Zoo | 0.00 | 0.00 | 0.42 | 0.00 | 0.21 | 0.00 | 0.21 | 0.42 | 1.25 | 0.21 | 0.00 | 0.25 | 60.42 |
| Qwen-2.5-7B-SimpleRL-Zoo | 6.25 | 7.29 | 6.25 | 7.29 | 8.33 | 3.75 | 5.42 | 2.08 | 5.42 | 4.38 | 2.50 | 5.36 | 77.16 |
| Qwen-2.5-14B-SimpleRL-Zoo | 12.71 | 13.13 | 13.13 | 10.42 | 13.33 | 9.17 | 9.79 | 3.54 | 10.42 | 11.46 | 5.63 | 10.25 | 0.42 |
| Qwen-2.5-Math-7B-SimpleRL-Zoo | 25.83 | 15.42 | 13.96 | 11.25 | 13.33 | 5.00 | 8.13 | 1.67 | 10.21 | 8.54 | 2.50 | 10.53 | 11.29 |
| Qwen2.5-Math-7B-Oat-Zero | 28.33 | 12.92 | 5.42 | 8.54 | 11.67 | 6.25 | 8.75 | 1.04 | 12.29 | 6.67 | 0.42 | 9.30 | 20.57 |
| s1.1-7B | 14.38 | 10.42 | 10.42 | 10.21 | 11.46 | 5.21 | 7.92 | 1.67 | 8.75 | 7.71 | 0.21 | 8.03 | 7.95 |
| DAPO-Qwen-32B | 54.58 | 50.00 | 51.67 | 50.00 | 50.00 | 42.50 | 36.04 | 19.17 | 40.83 | 45.42 | 27.29 | 42.14 | 5.91 |
| OpenThinker2-32B | 74.17 | 61.88 | 55.42 | 56.67 | 55.42 | 59.17 | 49.17 | 13.96 | 56.88 | 37.71 | 34.58 | 50.45 | 22.08 |
| S1.1-32B | 58.75 | 55.21 | 49.17 | 51.25 | 53.33 | 36.04 | 41.25 | 19.38 | 44.58 | 46.88 | 17.08 | 42.99 | 7.80 |
| *Multilingual AIME25* | | | | | | | | | | | | | |
| DeepSeek-R1-Distill-Qwen-7B | 29.58 | 20.21 | 21.25 | 22.29 | 19.79 | 5.63 | 8.75 | 0.42 | 26.67 | 10.00 | 0.00 | 14.96 | 6.97 |
| Open-Reasoner-Zero-7B | 14.58 | 13.33 | 11.88 | 9.58 | 11.04 | 1.67 | 9.38 | 0.00 | 10.21 | 9.79 | 0.21 | 8.33 | 10.04 |
| OpenThinker2-7B | 28.33 | 21.67 | 20.63 | 17.08 | 21.46 | 9.38 | 17.08 | 2.71 | 25.00 | 24.38 | 2.08 | 17.25 | 39.77 |
| OpenThinker3-7B | 22.50 | 27.71 | 23.33 | 20.00 | 27.50 | 6.67 | 14.79 | 3.13 | 28.54 | 21.67 | 1.67 | 17.95 | 63.28 |
| Qwen-2.5-1.5B-SimpleRL-Zoo | 0.00 | 0.00 | 0.00 | 0.00 | 0.00 | 0.00 | 0.00 | 0.00 | 0.21 | 0.00 | 0.00 | 0.02 | 61.14 |
| Qwen-2.5-7B-SimpleRL-Zoo | 4.58 | 3.33 | 2.08 | 3.96 | 5.42 | 0.83 | 3.13 | 0.63 | 1.46 | 2.50 | 0.21 | 2.56 | 79.51 |
| Qwen-2.5-14B-SimpleRL-Zoo | 13.96 | 11.67 | 10.42 | 10.83 | 10.00 | 3.54 | 6.46 | 1.88 | 6.67 | 8.54 | 2.08 | 7.82 | 0.61 |
| Qwen-2.5-Math-7B-SimpleRL-Zoo | 13.75 | 9.58 | 6.04 | 5.42 | 9.79 | 2.71 | 5.63 | 1.04 | 6.25 | 3.75 | 1.04 | 5.91 | 12.12 |
| Qwen2.5-Math-7B-Oat-Zero | 10.00 | 9.38 | 2.29 | 6.67 | 4.38 | 1.46 | 2.08 | 1.04 | 6.67 | 2.92 | 0.42 | 4.30 | 22.12 |
| s1.1-7B | 13.96 | 11.67 | 9.58 | 6.88 | 11.88 | 2.08 | 7.08 | 0.21 | 9.79 | 5.21 | 0.00 | 7.12 | 6.17 |
| DAPO-Qwen-32B | 38.13 | 38.54 | 37.29 | 36.25 | 34.58 | 30.83 | 32.71 | 18.33 | 31.67 | 34.17 | 22.29 | 32.25 | 4.56 |
| OpenThinker2-32B | 57.29 | 50.00 | 48.13 | 52.29 | 43.96 | 45.42 | 41.88 | 12.50 | 52.50 | 36.04 | 25.63 | 42.33 | 22.65 |
| S1.1-32B | 50.00 | 43.54 | 38.33 | 43.33 | 42.71 | 29.38 | 31.88 | 16.04 | 38.75 | 35.42 | 14.58 | 34.91 | 8.05 |
| *Multilingual GPQA-Diamond* | | | | | | | | | | | | | |
| DeepSeek-R1-Distill-Qwen-7B | 32.32 | 33.33 | 33.33 | 35.35 | 35.35 | 18.18 | 21.21 | 23.74 | 29.80 | 14.65 | 14.14 | 26.49 | 6.11 |
| Open-Reasoner-Zero-7B | 37.37 | 26.77 | 31.82 | 33.33 | 33.33 | 24.24 | 32.83 | 12.63 | 33.33 | 26.77 | 7.07 | 27.23 | 6.20 |
| OpenThinker2-7B | 28.79 | 17.68 | 17.17 | 16.67 | 22.73 | 22.22 | 25.76 | 14.14 | 22.22 | 18.69 | 14.14 | 20.02 | 38.15 |
| OpenThinker3-7B | 23.23 | 18.69 | 22.22 | 16.67 | 24.24 | 5.05 | 14.65 | 12.63 | 21.72 | 10.10 | 7.07 | 16.02 | 59.23 |
| Qwen-2.5-1.5B-SimpleRL-Zoo | 20.71 | 15.66 | 23.74 | 16.67 | 24.75 | 10.10 | 18.18 | 13.13 | 17.17 | 21.21 | 8.59 | 17.26 | 14.69 |
| Qwen-2.5-7B-SimpleRL-Zoo | 30.30 | 31.82 | 29.80 | 31.82 | 33.84 | 20.71 | 23.74 | 17.17 | 29.80 | 23.74 | 10.10 | 25.71 | 3.49 |
| Qwen-2.5-14B-SimpleRL-Zoo | 41.92 | 40.40 | 34.85 | 40.91 | 39.39 | 27.78 | 34.85 | 29.80 | 39.39 | 33.33 | 26.26 | 35.35 | 2.62 |
| Qwen-2.5-Math-7B-SimpleRL-Zoo | 30.81 | 26.26 | 22.73 | 27.78 | 28.28 | 18.18 | 17.17 | 9.09 | 27.27 | 16.67 | 8.08 | 21.12 | 12.26 |
| Qwen2.5-Math-7B-Oat-Zero | 25.76 | 17.17 | 7.07 | 21.21 | 15.66 | 19.19 | 21.72 | 9.09 | 30.81 | 6.06 | 12.63 | 16.94 | 19.74 |
| s1.1-7B | 17.68 | 14.14 | 20.20 | 22.22 | 29.29 | 9.09 | 17.17 | 16.16 | 24.75 | 16.67 | 18.69 | 18.73 | 11.98 |
| DAPO-Qwen-32B | 52.50 | 44.44 | 40.91 | 48.99 | 41.92 | 37.37 | 42.93 | 31.82 | 46.97 | 47.98 | 30.81 | 42.42 | 5.88 |
| OpenThinker2-32B | 62.63 | 57.58 | 58.08 | 59.09 | 58.59 | 50.51 | 47.47 | 21.72 | 56.57 | 0.00 | 0.00 | 42.93 | 22.91 |
| S1.1-32B | 64.65 | 57.58 | 57.58 | 59.60 | 56.57 | 41.41 | 48.48 | 36.36 | 56.57 | 53.03 | 32.83 | 51.33 | 11.85 |

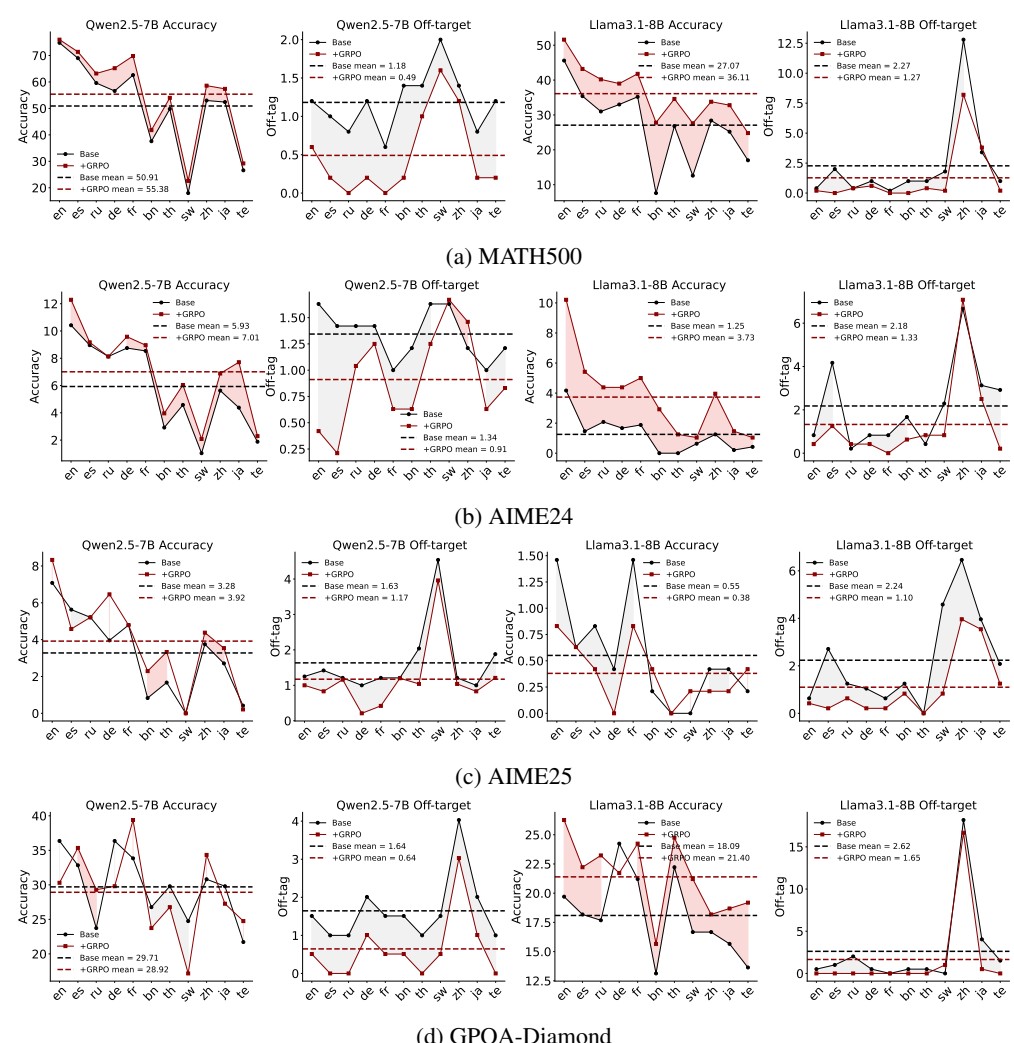

(a) MATH500

(b) AIME24

(c) AIME25

(d) GPQA-Diamond

Figure 9: **The Impact of Different Model Families in Interventional Study.** Multilingual reasoning performance across languages, comparing the influence of model family using Qwen2.5-7B-Instruct and Llama3.1-8B-Instruct as initial models. *"Base"* represents the performance of the initial model, while *"+GRPO"* denotes performance after fine-tuning with GRPO on English data. The light red area denotes the improvement in accuracy between the "Base" and "+GRPO" models, while the light gray area represents the reduction in the off-target rate between the two.

### E.3.2 THE DETAILED RESULTS IN PARALLEL SCALING LAW

Table 9 presents the detailed accuracy across languages with different numbers of parallel languages in Parallel Scaling Law. Table 10 presents the multilingual transfer metrics across languages with different numbers of parallel languages in Parallel Scaling Law.

### E.3.3 SCALING LAW INTERPRETATION: THE DRIVERS BEHIND THE EXPONENTS

We argue that the sublinear exponents in our power-law fits for accuracy and transferability arise from the principle of diminishing returns in the model's progression toward a unified, language-agnostic representation.

The very low exponent for accuracy ($\beta = 0.02$) indicates that reasoning performance is not primarily constrained by a lack of multilingual exposure, but rather by the intrinsic difficulty of the reasoning task itself. Since large language models are already pre-trained on massive corpora, they

Table 7: **The Impact of Model Size in Interventional Study.** Δ Performance on various benchmarks across *Qwen2.5-1.5B-Instruct* and *Qwen2.5-7B-Instruct*.

| Settings | Δ Performance | | | | | | | | | | | Average across languages | |
|---|---|---|---|---|---|---|---|---|---|---|---|---|---|
| | en | es | ru | de | fr | bn | th | sw | zh | ja | te | *Training* | *Untraining* |
| *Qwen2.5-1.5B-Instruct with GRPO on En Data* | | | | | | | | | | | | | |
| MATH500 | 20.40 | 27.20 | 17.40 | 14.40 | 17.40 | 6.20 | 7.80 | 4.40 | 16.80 | 14.20 | 7.40 | 20.40 | 13.32 |
| AIME24 | 2.08 | 0.42 | -0.21 | 1.25 | 0.42 | 0.21 | 0.21 | 0.00 | -0.21 | 0.21 | 0.21 | 2.08 | 0.25 |
| AIME25 | 1.04 | 0.21 | 0.00 | 0.21 | 0.00 | 0.00 | 0.00 | -0.21 | 0.42 | 0.21 | 0.21 | 1.04 | 0.10 |
| GPQA-Diamond | 9.09 | 20.71 | -0.51 | 5.05 | 12.12 | 1.52 | -2.53 | 2.02 | 8.59 | 3.54 | 0.51 | 9.09 | 5.10 |
| *Qwen2.5-7B-Instruct with GRPO on En Data* | | | | | | | | | | | | | |
| MATH500 | 4.40 | 1.00 | 1.40 | 5.60 | 5.60 | 4.20 | 5.60 | 1.40 | 0.40 | 6.20 | 0.80 | 4.40 | 3.22 |
| AIME24 | 2.71 | 1.04 | 0.83 | 1.46 | 0.42 | 0.42 | 0.42 | 2.08 | 2.08 | 4.58 | 0.21 | 2.71 | 1.35 |
| AIME25 | 1.25 | -1.04 | 0.00 | 2.50 | 0.00 | 1.46 | 1.67 | 0.00 | 0.63 | 0.83 | -0.21 | 1.25 | 0.58 |
| GPQA-Diamond | -3.54 | 4.55 | 0.00 | -1.01 | 7.07 | -2.02 | -1.01 | -7.07 | 4.04 | -3.03 | 0.51 | -3.54 | 0.20 |

Table 8: **The Language Settings in Parallel Scaling Law.**

| Settings | Training Parallel Languages |
|---|---|
| Only English | *en* |
| *w.* One parallel | *en, ru* |
| *w.* Two parallel | *en, ru, fr* |
| *w.* Three parallel | *en, ru, fr, es* |
| *w.* Four parallel | *en, ru, fr, es, de* |
| *w.* Five parallel | *en, ru, fr, es, de, bn* |
| *w.* Six parallel | *en, ru, fr, es, de, bn, th* |
| *w.* Seven parallel | *en, ru, fr, es, de, bn, th, zh* |

Table 9: **The Detailed Results in Parallel Scaling Law. Part 1:** Accuracy across languages with different numbers of parallel languages.

| Settings | Accuracy per language | | | | | | | | | | | Average | |
|---|---|---|---|---|---|---|---|---|---|---|---|---|---|
| | en | es | ru | de | fr | bn | th | sw | zh | ja | te | Acc | Off-tag |
| *Multilingual MATH500* | | | | | | | | | | | | | |
| Only English | 79.2 | 70.0 | 61.0 | 62.2 | 68.2 | 41.8 | 55.4 | 53.4 | 19.4 | 58.6 | 27.4 | 54.2 | 0.5 |
| *w.* One parallel | 78.4 | 73.4 | 66.0 | 65.6 | 67.2 | 48.8 | 57.4 | 57.8 | 26.2 | 62.0 | 33.8 | 57.9 | 0.2 |
| *w.* Two parallel | 79.0 | 73.4 | 64.4 | 67.4 | 69.2 | 45.2 | 60.2 | 63.0 | 26.2 | 61.6 | 32.6 | 58.4 | 0.2 |
| *w.* Three parallel | 77.8 | 73.6 | 64.6 | 68.4 | 69.8 | 46.0 | 60.8 | 62.2 | 24.0 | 60.8 | 34.2 | 58.4 | 0.4 |
| *w.* Four parallel | 77.2 | 71.2 | 66.2 | 66.8 | 68.0 | 47.6 | 61.8 | 62.0 | 28.6 | 60.2 | 35.2 | 58.6 | 0.6 |
| *w.* Five parallel | 77.4 | 71.4 | 62.2 | 66.2 | 66.0 | 48.6 | 62.0 | 62.2 | 32.4 | 63.8 | 37.0 | 59.0 | 0.4 |
| *w.* Six parallel | 76.4 | 70.8 | 63.8 | 65.6 | 66.8 | 48.6 | 61.8 | 34.6 | 63.4 | 63.4 | 38.4 | 59.4 | 0.5 |
| *w.* Seven parallel | 76.6 | 71.2 | 63.6 | 66.2 | 66.2 | 49.4 | 62.6 | 33.8 | 63.5 | 63.4 | 38.2 | 59.5 | 0.2 |

possess strong logical foundations and broad factual knowledge. Parallel training mainly helps refine how this existing knowledge is applied across languages, rather than imparting fundamentally new reasoning abilities. As a result, the incremental accuracy gains from each additional language remain marginal.

By contrast, the much higher exponent for transferability ($\beta = 0.29$) represents the central finding of our study. This value reflects that the main advantage of parallel training lies not in boosting raw accuracy but in reshaping the model's internal mechanisms. Specifically, it signals the emergence of a "learning-to-learn" skill: the ability to abstract away from language-specific surface patterns and consolidate a more robust cross-lingual representation. While each added parallel language

Table 10: **The Detailed Results in Parallel Scaling Law. Part 2:** Relative gain across languages with varying numbers of parallel training languages. $\Delta R_{\text{train}}$ and $\Delta R_{\text{target}}$ denote relative gains on training and target languages, respectively. MTI indicates multilingual transfer index.

| Settings | Relative Gain | | | | | | | | | | | Transfer Metrics | | |
|---|---|---|---|---|---|---|---|---|---|---|---|---|---|---|
| | en | es | ru | de | fr | bn | th | sw | zh | ja | te | $\Delta R_{\text{train}}$ | $\Delta R_{\text{target}}$ | MTI |
| *Multilingual MATH500* | | | | | | | | | | | | | | |
| Only English | 0.059 | 0.014 | 0.023 | 0.099 | 0.089 | 0.112 | 0.112 | 0.078 | 0.008 | 0.118 | 0.030 | 0.059 | 0.068 | 1.163 |
| *w.* One parallel | 0.048 | 0.064 | 0.107 | 0.159 | 0.073 | 0.298 | 0.153 | 0.456 | 0.091 | 0.183 | 0.271 | 0.078 | 0.194 | 2.496 |
| *w.* Two parallel | 0.056 | 0.064 | 0.081 | 0.191 | 0.105 | 0.202 | 0.209 | 0.456 | 0.189 | 0.176 | 0.226 | 0.081 | 0.214 | 2.650 |
| *w.* Three parallel | 0.040 | 0.067 | 0.084 | 0.208 | 0.115 | 0.223 | 0.221 | 0.333 | 0.174 | 0.160 | 0.286 | 0.076 | 0.229 | 3.002 |
| *w.* Four parallel | 0.024 | 0.032 | 0.121 | 0.180 | 0.070 | 0.266 | 0.241 | 0.644 | 0.170 | 0.149 | 0.323 | 0.088 | 0.290 | 3.282 |
| *w.* Five parallel | 0.008 | 0.049 | 0.047 | 0.201 | 0.102 | 0.319 | 0.209 | 0.633 | 0.174 | 0.218 | 0.391 | 0.105 | 0.365 | 3.475 |
| *w.* Six parallel | 0.021 | 0.026 | 0.070 | 0.159 | 0.067 | 0.293 | 0.241 | 0.922 | 0.196 | 0.210 | 0.444 | 0.125 | 0.443 | 3.534 |
| *w.* Seven parallel | 0.024 | 0.032 | 0.067 | 0.170 | 0.058 | 0.314 | 0.257 | 0.878 | 0.198 | 0.210 | 0.436 | 0.140 | 0.508 | 3.631 |

strengthens this capacity, the marginal benefit diminishes as the representation stabilizes, naturally producing a sublinear scaling curve.

**Theoretical Intuition** The emergence of the Parallel Scaling Law can be understood through the intuitive principle of diminishing returns in learning abstract representations. When a model is fine-tuned with only one or two parallel languages, it is forced to move beyond language-specific surface features and begin to form a language-agnostic, unified representation for reasoning. This initial shift is highly impactful and yields a disproportionately large gain in performance and transferability, which perfectly explains the First-Parallel Leap. As more parallel languages are added, the model's core mechanism for cross-lingual abstraction becomes increasingly robust. At this point, each additional language contributes less and less marginal information, as the model has already mastered the core skill of mapping reasoning concepts across languages.

### E.3.4 THE IMPACT OF SELECTED LANGUAGES

Figures 10 present a detailed analysis of accuracy and relative gain across a selection of parallel languages. As shown in Figure 10a, 10b and 10c, the relative gain on low-resource languages (*bn*, *sw*, and *te*) is consistently the largest, regardless of the chosen parallel language. In contrast, for high-resource languages (*ru*, *de*, and *zh*), the model's accuracy remains comparable across all settings of parallel training. A notable exception arises for *bn*: when trained with *bn* as the parallel language, accuracy on *bn* improves substantially compared to training with any other language.

Figure 10d presents the accuracy and relative gain on GPQA-Diamond. We observe that *ru* achieves the largest relative gain. This is because, as shown in Table 4, Qwen2.5-7B-Instruct performs relatively poorly on *ru* in GPQA compared to other high-resource languages, thereby yielding a larger relative gain.

These results suggest that **while low-resource languages consistently benefit the most from parallel training, and certain languages (e.g., *bn* and *ru*) exhibit language-specific effects, the overall outcomes of parallel training are largely robust to the choice of the parallel language.**

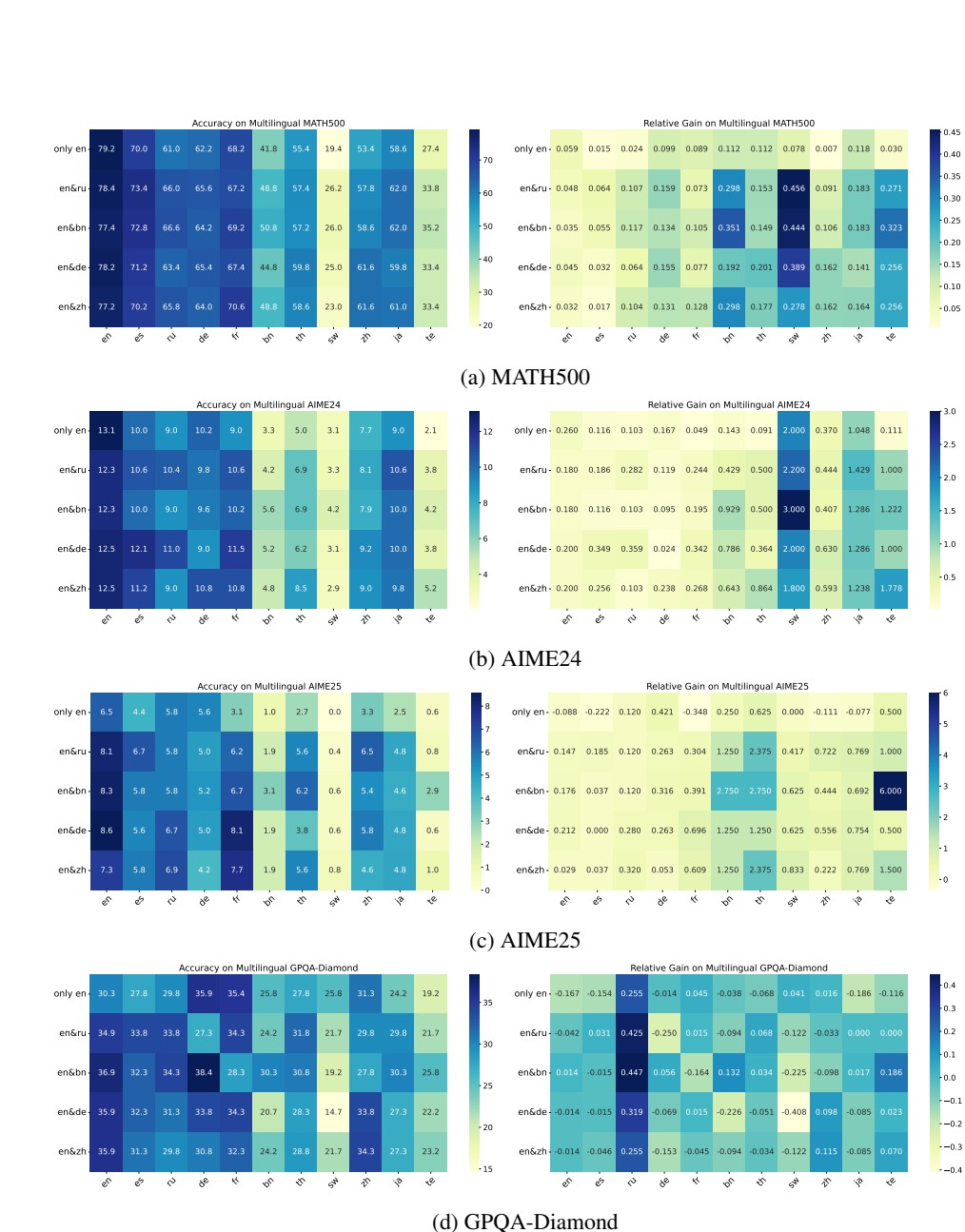

Figure 10: **The Analysis of Parallel Scaling Law across Selected Parallel Languages.** The accuracy and relative gain across various benchmarks with different parallel languages. "Only en" denotes only fine-tuned on English data. "en&LANGUAGE" indicates the model was fine-tuned on English and a parallel language, with LANGUAGE representing *ru, bn, de, zh*, respectively.

# F  Prompts Template

## F.1  Multilingual Reasoning Instruction

> **The Instruction Used in Multilingual Reasoning Prompt**
>
> Please always think in `[LANGUAGE]`.
>
> Solve the following mathematics problem step by step. At the end, provide your final answer enclosed in `\boxed{}`.
>
> Problem: {}

## F.2  Prompt hacking to force response language

> **The Prefixes Used in Prompt Hacking.** Note that We list seven out of eleven languages.
>
> - **English:** By request, I will start thinking in English.
> - **Japanese:** 要求があれば、日本語で考え始めます。
> - **Chinese:** 应要求，我将开始用中文思考。
> - **Spanish:** A petición, empezaré a pensar en español.
> - **French:** Sur demande, je commencerai à penser en français.
> - **German:** Auf Anfrage werde ich anfangen, in Deutsch zu denken.
> - **Swahili:** Kwa ombi, nitaanza kufikiria kwa Kiswahili.

## F.3  Template for R1-like Reasoning

> **The Template for R1-like Reasoning**
>
> You are a helpful AI Assistant that provides well-reasoned and detailed responses. You first think about the reasoning process as an internal monologue and then provide the user with the answer. The final answer must be put in `\boxed{}`. Respond in the following format: `<think>\n...\n</think>\n<answer>\n...\n</answer>`

