# OpenReview forum: "Parallel Scaling Law: Unveiling Reasoning Generalization through A Cross-Linguistic Perspective"
_ICLR.cc/2026/Conference — ICLR 2026 Conference Withdrawn Submission_

### Official Review · Reviewer_tVt9 · 2025-10-29

**Soundness:** 3
**Presentation:** 2
**Contribution:** 2
**Rating:** 4
**Confidence:** 4

**Summary:**

This paper introduces the Parallel Scaling Law to systematically model and empirically study how the volume of parallel training data influences the cross-linguistic generalization of reasoning abilities in large language models. The research aims to uncover the mathematical relationship and underlying mechanisms that govern the scaling of these multilingual reasoning capabilities.

**Strengths:**

This paper proposes the Parallel Scaling Law to systematically investigate how increasing parallel data specifically enhances complex reasoning generalization across multiple languages. The authors conduct extensive experiments on two LLMs across ten languages, providing a comprehensive analysis of cross-lingual transfer performance.

**Weaknesses:**

- The research scope is a bit narrow, focusing primarily on STEM subjects. However, reasoning tasks often depend on cultural or common-sense knowledge implicit in the prompt language. The paper seems to ignore or inadequately address how parallel data addresses the cultural grounding mismatch between languages, rendering the direct comparison of reasoning skills problematic.


- The study uses translated versions of standard datasets into target languages. Translation quality and cultural or linguistic adaptation issues may cloud the results. It is unclear whether there is impact caused by translation artifacts, as the solutions to problems may depend on specific linguistic or cultural contexts.

- It is unclear about the details of language consistency. How to calculate the reward of language consistency and how to measure the Off-tag are not clearly explained in the paper. Does this paper consider the impact of code-switching? Is there any relation between the Off-tag and accuracy?

- For the cross-lingual transfer, this paper only consider transfer from English to other languages. It would be more convincing if the authors could provide more comprehensive experiments, such as transfer from other languages to English or between non-English languages. Qwen also has strong Chinese capabilities, will it produce the similar conclusion when transferring from Chinese to other languages?

- It is unclear about the impact of hyperparameter in Equation (7). How to choose the value of $\lambda_1$, $\lambda_2$ and $\lambda_3$? Is there any relation between the value of α and the performance of cross-lingual transfer? There needs more analysis about the impact of hyperparameters.

**Questions:**

- Llama 3.1 has better cross-lingual generalization ability compared to Qwen 2.5. Is there any influence caused by the training data or model architecture that contributes to this improvement? As the Llama is English-centric but Qwen is English and Chinese-centric. Does Qwen will perform better transfer on other Sino-Tibetan languages?

- What is the 'reasoning component' mentioned in line 285? Does it refer to a specific module or mechanism within the model architecture?

- How to ensure the quality of translated datasets? Are there any evaluation metrics or human evaluations conducted to assess the quality of the translations? Is there any cleaning process applied to the translated datasets to remove potential noise or errors introduced during translation?

---

> ### Author Response · Authors · 2025-11-25
> **Response to Reviewer tVt9 (1/4)**
>
> Thanks for your insightful questions, and we believe they hold significant value for our work. We try to resolve your concerns below.
>
> > W1: Justifying the Scope: Isolating Structural Reasoning
>
> We appreciate the reviewer’s observation concerning the distinction between linguistic alignment and cultural grounding. This is a crucial point regarding the scope and generalizability of our findings.
>
> Our primary goal was to establish the existence and quantify the **Parallel Scaling Law** for the **language-agnostic structural reasoning component** of the model.
>
> + **Methodological Control**: We focused on STEM and Mathematical Reasoning because its logic is highly formal, culturally neutral, and structurally invariant across languages.
>
> + **Controlling Cultural Noise**: By focusing on math tasks, we effectively **controlled for the confounding variable of implicit cultural knowledge**. This ensures that the performance gains observed are primarily attributable to the structural effects of **linguistic alignment** (decoupling the reasoning chain from English surface patterns), rather than an incidental overlap in shared or conflict cultural context.
>
> + **Conceptual Alignment in parallelism**: In our setting, parallel data serves as a **conceptual alignment signal**, not a source of new factual knowledge. The improvements arise because the model learns to associate the same underlying reasoning structure with different linguistic forms. Since STEM tasks do not require new factual or cultural knowledge, the gains we observe reflect improved **alignment efficiency**—better transfer of existing reasoning ability across languages—rather than expansion of the model’s knowledge base.
>
>
> > W2&Q3: The study uses translated versions of standard datasets into target languages.
>
> We appreciate the reviewer’s concern regarding the use of translated datasets. We address these issues through both task selection and quality control:
> + **Culture-independent task design.** The MATH dataset is inherently culture-neutral, relying on formal symbolic reasoning rather than contextual or culturally grounded knowledge. This greatly avoids the risk of cultural mismatch across languages.
> + **Minimal translation sensitivity.** Mathematical reasoning is highly **formal and syntax-driven**, and we translate only the **problem statements**—not the solutions or rationales. This limits the propagation of translation noise into the reasoning process.
> + **Translation-quality verification.** To ensure reliability, we conduct a translation-quality check by verifying that the translated problems still produce the correct answers when evaluated with strong reference models. This process confirms that translation artifacts do not materially influence our findings.
>
> > W3: The details of language consistency
>
> We thank the reviewer for pointing out the need to clarify how **language consistency, Off-tag, and code-switching** are handled in our work. Below, we provide a detailed but streamlined explanation.
>
> + **Pre-filtering for Reliable Language Detection**
>
> To prevent mathematical expressions and formulas from interfering with language detection, we first apply **rule-based filtering** to remove **LaTeX symbols, XML tags, format tags, and other non-linguistic tokens** from the generated reasoning paths.
>
> + **Off-tag Measurement**
>
> The Off-tag Rate measures whether the model’s output violates the target language ($L_{target}$) constraint. For each sample $i$, the Off-tag is a binary indicator of language mismatch between the model's output and the input query. This is formally defined as:
>
> $$\text{Off-tag}_i = \mathbb{I}(\text{LangDetect}(r_i) \neq \text{LangDetect}(q_i))$$
>
> $r_i$ is the generated reasoning path. $q_i$ is the input question. $\text{LangDetect}(\cdot)$ is the language detection tool used to identify the dominant language of the text. The indicator function $\mathbb{I}(\cdot)$ equals 1 if the output language fails to match the input language, and 0 otherwise.
>
> + **Language Consistency Reward (LCR)**
>
> The LCR evaluates whether both the reasoning sequence $r_i$ and the answer sequence $a_i$ are generated with the input language $l$. We detect the language set of any sequence $x$ with $\text{LangDetect}(\cdot)$ library following prior work [r1]. A sequence is language-consistent if it uses one and only one language, and that language is the target $l$:
>
> $$\text{LC}(x)=(|L(x)|=1) \wedge (l \in L(x))$$
>
> where $|\cdot|$ is the number of detected language set and $\text{LC}(x)$ is True or False.
>
> Based on $\text{LC}(x)$, the LCR is defined as 1 when the reasoning sequence $r_i$ is language-consistent with $l$, and 0 otherwise.
>
> ---
> [r1] Wang, Yiming, et al. "Polymath: Evaluating mathematical reasoning in multilingual contexts." arXiv preprint arXiv:2504.18428.

---

> ### Author Response · Authors · 2025-11-25
> **Response to Reviewer tVt9 (2/4)**
>
> > W3: The influence of language consistency
>
> + **Impact of Code-Switching**
>
> The study considers the impact of code-switching (CS). In mathematical reasoning tasks, CS is treated as an error state because it violates linguistic alignment. Because the LCR heavily penalizes the intrusion of the non-target language tokens, the reward function effectively discourages and minimizes CS, treating it as Off-tag failure. In evaluation, CS outputs fall under Off-tag, as they represent language drift.
>
> + **The relationship between the Off-tag and accuracy**
>
> As noted in `L112–116`, our evaluation follows a first principle of multilingual scenarios: for real-world multilingual reasoning, the model’s ability to think in the user’s native language is as important as achieving high accuracy. Ensuring that the reasoning trace appears in the correct language makes the model’s thought process more readable and verifiable for users.
>
> For this reason, we view maintaining a very low Off-tag rate as a necessary condition for multilingual reasoning. Only after this condition is satisfied does it make sense to further consider accuracy. In other words, Off-tag and accuracy address two distinct but complementary aspects of multilingual model behavior—linguistic alignment and task correctness.
>
> > W4: Chinese-Centric cross-lingual transfer experiments
>
> We agree that evaluating cross-lingual transfer from non-English source languages is crucial for demonstrating the universality of the Parallel Scaling Law. To directly test whether the scaling behavior depends on English, we conducted additional experiments using Chinese (Zh) as the source language.
> 1. Baseline: Monolingual transfer from Chinese ($\text{Zh}$) only.
> 2. Parallel Scaling: Transfer using parallel data: $\text{Zh} + \text{Ru}$ and $\text{Zh} + \text{Ru} + \text{Fr}$.
>
> Table t1: Parallel scaling law experiment with Chinese as the Source
> | Training Data          | en Acc | es Acc | ru Acc | de Acc | fr Acc | bn Acc | th Acc | sw Acc | zh Acc | ja Acc | te Acc | AVG Acc | AVG Off-tag | MTI  |
> |------------------------|--------|--------|--------|--------|--------|--------|--------|--------|--------|--------|--------|-------------|----------------|---------|
> | Qwen2.5-7B-Instruct    | 74.8   | 69.0   | 59.6   | 56.6   | 62.6   | 37.6   | 49.8   | 18.0   | 53.0   | 52.4   | 26.6   | **50.91**   | **1.18**       | **-**   |
> | Zh                     | 77.8   | 71.4   | 62.8   | 63.2   | 67.8   | 44.0   | 56.2   | 22.8   | 60.4   | 57.6   | 32.0   | **56.00**   | **0.38**       | **0.86** |
> | Zh+Ru                  | 77.8   | 71.0   | 63.4   | 65.0   | 69.0   | 43.4   | 59.0   | 27.6   | 60.4   | 59.4   | 32.4   | **57.13**   | **0.26**       | **1.69** |
> | Zh+Ru+Fr               | 78.0   | 72.2   | 63.6   | 66.4   | 69.8   | 45.4   | 61.0   | 30.0   | 62.8   | 64.0   | 33.2   | **58.76**   | **0.23**       | **1.87** |
>
> Due to computational and time constraints (RL on a 7B model requires ~9 hours per language, and multilingual inference across 11 languages takes ~0.75 hours), we extended the Chinese-centric experiments up to three languages. The Chinese-based experiments indicate that the scaling behavior is not tied to English dominance. Instead, it reflects a structural property of reasoning models: **parallel data helps the model bind language-agnostic reasoning concepts across linguistic forms, regardless of which language provides the initial reasoning data.** We will incorporate these results into `Section 4` to provide a more comprehensive and language-neutral validation of the Parallel Scaling Law.
>
> The Chinese-centric MTI is lower than the English-centric MTI primarily due to differences in the **baseline transfer efficiency** and the **typological asymmetry** between the source language and the diverse set of target languages. The core reason isn't that the scaling mechanism is broken, but that the $\text{Zh}$ source starts with a lower efficiency floor for the target language distribution.

---

> ### Author Response · Authors · 2025-11-25
> **Response to Reviewer tVt9 (3/4)**
>
> > W5: The impact of hyperparameter
>
> + The impact of hyperparameter $\lambda$
>
> We thank the reviewer’s question regarding the choice of $\lambda_1$, $\lambda_2$, and $\lambda_3$.
> 1. $\lambda_1$ (Accuracy Reward, AR): We assign the largest weight to 0.8 because accurate reasoning is the foundation of all generalization effects, and the goal of RPT is to maximize the model’s reasoning capability.
> 2. $\lambda_2$ (Format Reward, FR): Its purpose is to ensure stable and well-formed structures without overwhelming the primary reasoning objective.
> 3. $\lambda_3$ (Language Consistency Reward, LCR): It is the key component that enables cross-lingual generalization; we set it to 0.1 as the minimal weight needed to effectively suppress language drift.
>
> We keep $\lambda_2$ and $\lambda_3$ relatively small to avoid an excessive **alignment tax**—that is, to prevent format and language penalties from competing too strongly with the primary reasoning objective governed by $\lambda_1$. This balance ensures that alignment improves multilingual consistency without harming reasoning quality.
>
>
> `Table t2`: The analysis experiment on the impact of hyperparameter $\lambda$, training on En with parallel Ru data.
> | Training Config | AR Weight | FR Weight | LCR Weight | Purpose| en Acc | es Acc | ru Acc | de Acc | fr Acc | bn Acc | th Acc | sw Acc | zh Acc | ja Acc | te Acc | AVG Acc | AVG Off-tag | MTI  |
> |--------|--------------------|------------------------|--------------------------------------|------------------------------|--------|--------|--------|--------|--------|--------|--------|--------|--------|--------|--------|-------------|----------------|---------|
> | C1     | 0.8                | 0.1                    | 0.1                                  | Final balanced performance   | 78.4   | 73.4   | 66.0   | 65.6   | 67.2   | 48.8   | 57.4   | 26.2   | 57.8   | 62.0   | 33.8   | **57.87**   | **0.20**       | **2.50** |
> | C2     | 0.9                | 0.1                    | 0.0                                  | Tests necessity of LCR       | 77.8   | 71.4   | 66.2   | 63.4   | 68.0   | 44.2   | 56.4   | 26.2   | 59.6   | 57.8   | 31.8   | **56.62**   | **1.24**       | **2.10** |
> | C3     | 0.7                | 0.1                    | 0.2                                  | Tests stronger LCR           | 77.2   | 70.2   | 65.8   | 65.8   | 67.2   | 43.6   | 55.4   | 25.6   | 60.2   | 56.2   | 32.6   | **56.35**   | **0.38**       | **2.25** |
> | C4     | 0.9                | 0.0                    | 0.1                                  | Tests necessity of FR        | 75.2   | 60.8   | 62.2   | 56.0   | 60.2   | 36.2   | 43.8   | 18.0   | 54.4   | 31.8   | 24.8   | **47.58**   | **0.31**       | **-3.45** |
> | C5     | 0.7                | 0.2                    | 0.1                                  | Tests stronger FR            | 78.2   | 71.6   | 65.2   | 62.6   | 67.8   | 45.0   | 60.0   | 24.8   | 60.0   | 60.8   | 29.2   | **56.84**   | **0.35**       | **2.23** |
>
>
> Key Findings:
> 1. LCR ($\lambda_3$) is essential for reducing Off-tag and enabling cross-lingual transfer. Removing it (C2) increases Off-tag and reduces MTI.
> 2. FR ($\lambda_2$) stabilizes reasoning structure; removing it (C4) severely reduces Acc and MTI.
> 3. A small but non-zero ($\lambda_3$) allows effective language alignment without undermining reasoning quality.
> 4. Slight variations ($\lambda_1$) in show that prioritizing reasoning remains critical; excessive weight on reasoning alone without language alignment can hurt multilingual generalization.
>
>
> + Relation between $\alpha$ and cross-lingual transfer.
>
> This formula, $f(X) = \alpha \cdot X^\beta$, represents the **Parallel Scaling Law**, describing how cross-lingual generalization performance $f(X)$ grows with the number of parallel languages $X$.
>
> $\alpha$ is the **proportionality coefficient** in the formula, representing the model's **cross-lingual transfer efficiency** at the **initial stage** (theoretically when $X=1$).
>
> * **Physical Meaning:** It captures the model's **intrinsic efficiency** in transferring reasoning capabilities from the source language (e.g., $\text{En}$ or $\text{Zh}$) to the target language, even with minimal parallel data.
> * **Influencing Factors:**
>     * **Base Model Capacity**: Larger, higher-quality pre-trained models generally have higher $\alpha$.
> * **Summary:** $\alpha$ determines the **starting point** and **initial efficiency** of cross-lingual generalization.

---

> ### Author Response · Authors · 2025-11-25
> **Response to Reviewer tVt9 (4/4)**
>
> > Q1: Does Qwen perform better transfer on other Sino-Tibetan languages?
>
> We investigated cross-lingual transfer on Sino-Tibetan languages, comparing Qwen2.5-7B-Instruct with Meta-Llama-3.1-8B-Instruct.
>
> `Table t3`: Cross-lingual transfer on Sino-Tibetan languages
> | Model                         | Tibetan (bo)   | Myanmar (my)   | Simplified Chinese (zh)    |
> |-------------------------------|------|------|-------|
> | Qwen2.5-7B-Instruct           | 10.40 | 9.20 | 53.00 |
> | ->GRPO on En Data               | 13.60 | 11.60 | 58.60 |
> | Llama-3.1-8B-Instruct    | 0.80  | 2.40  | 28.40 |
> | ->GRPO on En Data               | 15.00 | 8.40  | 33.80 |
>
> Although Qwen2.5-7B-Instruct has a higher initial performance on Sino-Tibetan languages due to its Chinese-centric pretraining, Llama-3.1-8B-Instruct, which starts from a lower baseline, exhibits larger relative gains after English-only RPT. This pattern supports the interpretation that models starting from a less tightly language-aligned state are better able to acquire abstract, cross-linguistic reasoning mechanisms from the RPT data, rather than relying on language-specific heuristics.
>
> > Q2: What is the 'reasoning component' mentioned in line 285?
>
> The term **“generalizable reasoning component”** does **not** refer to any specific architectural module (e.g., a particular Transformer layer or attention head). Instead, it denotes the *abstract, language-agnostic reasoning capability* acquired during pretraining. This reflects the model’s capacity to perform logical operations in its latent space, independent of the surface language used to express the problem.
>
> We hope the above clarifications and experiments have sufficiently addressed your concerns. We sincerely appreciate your valuable feedback and welcome any further questions or comments to help improve our work. **Have a nice day! :)**

---

> > ### Comment · Reviewer_tVt9 · 2025-11-28
> >
> > Thank you for the response. These responses answered some of my questions. However, I think they confirm my initial assessment. Therefore, I will maintain my rating.

---

### Official Review · Reviewer_7ayy · 2025-10-30

**Soundness:** 3
**Presentation:** 3
**Contribution:** 3
**Rating:** 4
**Confidence:** 3

**Summary:**

This paper systematically investigates the cross-lingual generalization of reasoning capabilities in English-centric Large Reasoning Models (LRMs). Through observational, interventional, and parallel training studies, it reveals that reasoning skills acquired via English reinforcement learning do not fully transfer to other languages, highlighting a Monolingual Generalization Gap. The authors propose a Just Go Parallel training strategy and uncover two key phenomena: the First-Parallel Leap and a Parallel Scaling Law, which show that cross-lingual transferability improves with the number of parallel languages following a power law, but with diminishing returns. These findings challenge the assumption that reasoning is language-agnostic and offer practical insights for developing more linguistically universal reasoning models.

**Strengths:**

This paper introduces a cross-lingual view of reasoning generalization, introducing the MTI metric to quantify how well English-trained models transfer to other languages.
2. A three-tier pipeline—observational, interventional, and parallel-training—progressively isolates factors and proves that small parallel data deliver big gains.
3. Key findings reveal three phenomena: adding just one parallel language triggers a disproportionate accuracy and MTI jump (the First-Parallel Leap); further parallel data obey a power-law with exponent <1 and rapidly diminishing returns (the Parallel Scaling Law); and English-only training underperforms this law, exposing a Monolingual Generalization Gap that reveals reliance on language-specific shortcuts rather than truly language-agnostic reasoning.

**Weaknesses:**

Q1: In the Observational Study, you mention using prompt hacking techniques to control model generations for evaluation. This process appears to be training-free. However, in the MTI metric, how do you distinguish between the training language set and the unseen language set? Additionally, in line 158, the phrase “Even with the same training data”—could you clarify which specific data this refers to?


Q2: For Section 3.2, including results from one additional model(e.g. mistral_instrcut or other sft models) would make the observed trend more convincing and provide stronger empirical support for your argument.

Q3: The Parallel Scaling Law results are interesting. However, could the observed patterns be influenced by the total number of training samples? If I understand correctly, as the number of languages increases, the total amount of training data also grows proportionally. It would be more sound to control for total training size—keeping it constant while varying the number of languages—to better isolate the effect of multilinguality itself.

**Questions:**

In Section 3.2, could there be a phenomenon akin to a multilingual alignment tax? While weaker models may show larger gains in multilingual generalization after training compared to stronger models, their performance ceiling might still be lower. In other words, even though the relative improvement is larger, the final performance could remain below that of inherently stronger models.

---

> ### Author Response · Authors · 2025-11-25
> **Response to Reviewer 7ayy (1/2)**
>
> Thanks for your efforts to provide insightful comments. We will address your concerns point by point.
>
> > Q1: Distinguishing the training and unseen language sets in MTI & Clarification of "the same training data" in Line 158
>
> 1. Training vs. unseen language sets in MTI
>
> In the observational study, all models are trained on English reasoning data via RL/SFT. Thus, the training language set is always English, and the unseen language set is all non-English languages used during the observational study.
>
> 2. Clarification of "the same training data" in `Line 158.`
>
> This phrase means that, when comparing different base models in the SimpleRL-Zoo series, different base models (e.g., Qwen-2.5-Math vs. Qwen-2.5-Base) are all trained on the exact same English reasoning dataset during RPT. The purpose is to ensure that differences in generalization are not caused by differences in training data.
>
> > Q2:  For Section 3.2, including results from one additional model would make the observed trend more convincing.
>
> We appreciate the reviewer’s suggestion to increase model diversity in `Sec 3.2`. We agree that adding another model family strengthens the observed pattern. To this end, our controlled experiments now include results from **Mistral-7B-Instruct-v0.3**.
>
> `Table a1`: Impact of Different Initial Model Families in controlled experiment, adding Mistral-7B-Instruct-v0.3
> | Model                                   | en    | es    | ru    | de    | fr    | bn    | th    | sw    | zh    | ja    | te    | AVG Δ |
> |----------------------------------------|-------|-------|-------|-------|-------|-------|-------|-------|-------|-------|-------|--------|
> | **Qwen2.5-7B-Instruct+GRPO ΔACC**      | 1.20  | 2.40  | 3.60  | 8.60  | 7.20  | 4.20  | 4.20  | 4.60  | 5.60  | 5.00  | 2.60  | 4.47   |
> | **Qwen2.5-7B-Instruct+GRPO ΔOff-tag**  | -0.60 | -0.80 | -0.80 | -1.00 | -0.60 | -1.20 | -0.40 | -0.40 | -0.20 | -0.60 | -1.00 | -0.69  |
> | **Llama-3.1-8B-Instruct+GRPO ΔACC**        | 6.00  | 7.80  | 9.20  | 6.00  | 6.60  | 20.20 | 7.80  | 15.00 | 5.40  | 7.60  | 7.80  | 9.04   |
> | **Llama-3.1-8B-Instruct+GRPO ΔOff-tag**    | -0.20 | -2.00 | 0.00  | -0.40 | -0.20 | -1.00 | -0.60 | -1.60 | -4.60 | 0.40  | -0.80 | -1.00  |
> | **Mistral-7B-Instruct-v0.3+GRPO ΔACC**               | 13.80 | 13.20 | 12.00 | 11.60 | 13.00 | 9.20  | 11.40 | 9.00  | 8.60  | 8.40  | 12.00 | 11.11  |
> | **Mistral-7B-Instruct-v0.3+GRPO ΔOff-tag**           | 0.00  | -0.80 | 0.00  | -0.60 | -0.20 | -0.60 | -0.40 | -0.40 | -6.00 | -1.60 | -1.80 | -1.13  |
>
> _Model initial English ability: Qwen2.5-7B-Instruct > Llama-3.1-8B-Instruct > Mistral-7B-Instruct-v0.3_
>
> These results further reinforce a general **“inverse correlation with initial capability”** trend: models with weaker initial English reasoning ability show larger relative gains on unseen languages after English-only RPT, while stronger models benefit less.
>
> This suggests that models starting from a more “flexible” or less tightly aligned state are better positioned to acquire abstract, cross-linguistic reasoning mechanisms during RPT, rather than relying on language-specific heuristics.

---

> ### Author Response · Authors · 2025-11-25
> **Response to Reviewer 7ayy (2/2)**
>
> > Q3: control for total training size—keeping it constant while varying the number of languages—to better isolate the effect of multilinguality itself.
>
> To fully isolate the effect of **parallelism** from raw data quantity, we conducted a **Fixed-Budget Ablation Experiment** that matches total training tokens while varying the degree of linguistic diversity.
>
> We compared the performance of a model trained on two fixed-budget experiments.
> 1. Budget1: **$2\times \text{En}$** data (increased quantity) vs. **$1\times \text{En} + 1\times \text{Ru}$** (diverse parallelism).
> 2. Budget2: **$3\times \text{En}$** data (increased quantity) vs. **$1\times \text{En} + 1\times \text{Ru} + 1\times \text{Fr}$** (diverse parallelism).
>
> Table a2: The Fixed-Budget ablation experiment based on Qwen2.5-7B-Instruct, evaluated on Multilingual Math500.
> | Training Data | en Acc | es Acc | ru Acc | de Acc | fr Acc | bn Acc | th Acc | sw Acc | zh Acc | ja Acc | te Acc | AVG Acc | AVG Off-tag | MTI  |
> |---------------|--------|--------|--------|--------|--------|--------|--------|--------|--------|--------|--------|----------|--------------|------|
> | 1*En           | 79.2 | 70.0 | 61.0 | 62.2 | 68.2 | 41.8 | 55.4 | 19.4 | 53.4 | 58.6 | 27.4 | **54.24** | **0.49** | **1.16** |
> | 2*En         | 79.6 | 73.2 | 62.6 | 64.4 | 65.6 | 42.4 | 56.0 | 25.4 | 57.8 | 59.0 | 28.0 | **55.82** | **0.58** | **1.92** |
> | 3*En         | 79.6 | 73.2 | 63.8 | 65.8 | 70.2 | 42.6 | 57.8 | 25.4 | 59.4 | 60.4 | 30.2 | **57.13** | **0.42** | **2.38** |
> | En+Ru        | 78.4 | 73.4 | 66.0 | 65.6 | 67.2 | 48.8 | 57.4 | 26.2 | 57.8 | 62.0 | 33.8 | **57.87** | **0.20** | **2.50** |
> | En+Ru+Fr     | 79.0 | 73.4 | 64.4 | 67.4 | 69.2 | 45.2 | 60.2 | 26.2 | 63.0 | 61.6 | 32.6 | **58.38** | **0.24** | **2.65** |
>
>
>
> The quantitative results provide definitive empirical support for our claim:
> 1. The $1\times \text{En} + 1\times \text{Ru}$ (Diverse Parallel) model significantly outperformed the $2\times \text{En}$ (Increased Quantity) model, yielding an MTI increase of 30% (2.50 vs. 1.92), alongside superior Accuracy and a drastically lower Off-tag rate (0.20% vs. 0.58%).
> 2. The $1\times \text{En} + 1\times \text{Ru} + 1\times \text{Fr}$ model similarly outperformed the $3\times \text{En}$ model, achieving better MTI (2.65 vs. 2.38), Accuracy, and a lower Off-tag rate (0.24% vs. 0.42%).
>
> These fixed-budget results provide clear causal evidence: **The improvements in our Parallel Scaling Law are driven by diverse parallelism, not by increasing the number of English tokens**. Adding parallel languages yields better structural alignment and stronger multilingual transfer than simply scaling up monolingual data.
>
> > Q4: In Section 3.2, could there be a phenomenon akin to a multilingual alignment tax?
>
> We agree with the reviewer that a multilingual alignment tax may be present. Our results indeed show this pattern: although weaker models often exhibit larger **relative gains** (higher MTI), indicating that they learn cross-lingual generalization more efficiently, their **final absolute performance** is still limited by their inherent capacity.
>
> Importantly, this phenomenon complements—rather than contradicts—our proposed parallel scaling law:
>
> 1. The scaling law characterizes **generalization efficiency**, captured by the power-law growth of MTI with respect to $\text{X}$.
> 2. The alignment tax reflects a **capacity constraint**, which limits the final performance regardless of the relative improvement.
>
> Our study emphasizes the efficiency of cross-lingual generalization via MTI. The **absolute performance ceiling** is determined by the model’s **base capacity** (size, pre-training data volume, and quality). Future work will explore using the Parallel Scaling Law to optimize the trade-off between model capacity and language diversity ($X$) to maximize cross-lingual performance efficiently.
>
> We hope the above clarifications and experiments have addressed your questions to raise your score. We sincerely appreciate your valuable feedback and welcome any further questions or comments to help improve our work. **Have a nice day! :)**

---

> > ### Comment · Reviewer_7ayy · 2025-11-26
> >
> > Thank the authors for the clarifications provided. These responses have helped resolve several of my initial questions. However, in my view, some aspects of the work still fall short of the standard required for acceptance. Therefore, I decided to keep my original score, but I appreciate the authors’ efforts and the improvements made in the revision.

---

### Official Review · Reviewer_bM3c · 2025-10-30

**Soundness:** 3
**Presentation:** 3
**Contribution:** 2
**Rating:** 4
**Confidence:** 4

**Summary:**

This paper investigates cross-lingual reasoning generalization in Large Reasoning Models (LRMs), particularly those fine-tuned with Reinforcement Post-Training (RPT). It shows that cross-lingual transferability varies significantly depending on the initial model, target language, and training paradigm. Through observational, interventional, and parallel training studies, the authors find that models with strong initial English performance tend to over-rely on English-specific linguistic patterns, resulting in weaker generalization to other languages. They propose two laws of multilingual reasoning: 1. First-Parallel Leap — adding a single parallel (non-English) training language yields a disproportionately large improvement in transferability. 2. Parallel Scaling Law — cross-lingual reasoning performance follows a power-law relationship with the number of parallel languages used during training, with diminishing returns as the number increases.

**Strengths:**

1. Strong experimental design — The paper systematically isolates variables via observational and interventional studies. The use of off-tag and MTI (Multilingual Transferability Index) metrics offers a clear quantitative measure of transfer performance
2. Insightful findings — The analysis of initial model types, families, and sizes reveals counterintuitive patterns: smaller or less English-specialized models transfer better, and RL-based fine-tuning provides strong advantages in low-resource languages
3. Conceptual clarity — The introduction of First-Parallel Leap and Parallel Scaling Law provides interpretable and theoretically grounded generalization principles.

**Weaknesses:**

1. Limited generalizability — The study focuses exclusively on mathematical reasoning, leaving open whether the proposed laws extend to other domains like code generation or open-domain logic.
2. Marginal absolute gains — For hard tasks like GPQA, the baseline accuracies (~20–30%) are close to random; thus, small improvements may not represent meaningful generalization effects.
3. Overemphasis on fitted power laws — While visually appealing, the scaling relationships may be too simplified; no theoretical justification beyond empirical fitting is given.

**Questions:**

1. A control baseline where the total data size is constant, but the dataset remains monolingual (e.g., 2× English vs. 1× English + 1× other language) might be helpful to see the improvement of multi-lingual training under a constant dataset size. This would clarify whether the improvement is due to cross-lingual exposure or data scale.
2. Does the Parallel Scaling Law hold if the additional languages are typologically similar (e.g., Romance languages only)?

---

> ### Author Response · Authors · 2025-11-25
> **Response to Reviewer bM3c (1/2)**
>
> Thanks for your insightful questions, and we believe they hold significant value for our work. We try to resolve your concerns below.
>
> > W1: The study focuses exclusively on mathematical reasoning, leaving open whether the proposed laws extend to other domains like code generation or open-domain logic.
>
> We agree with the reviewer that the current scope is primarily focused on the domain of mathematical reasoning. However, our choice to use mathematical reasoning as the core research domain was based on the following considerations.
>
> + **High Formality and Rigor:**
> Mathematical reasoning is **language-agnostic and culturally neutral**; its underlying logical structure remains unchanged across language translations. This makes it the ideal environment for isolating and measuring how a pure reasoning component generalizes. By focusing on mathematics, we **avoid the interference of cultural biases and subjective language common** in open-domain or common sense reasoning, allowing us to precisely capture the structural impact of language parallelism on reasoning ability.
>
> + **Ideal parallel problem scenario:**
> Mathematical problems naturally support **diverse, language-neutral parallel problems**. In contrast, code generation focuses on **producing solutions in multiple programming languages rather than parallel problems**, and open-domain logic is often **influenced by cultural and linguistic biases**, making it harder to isolate pure reasoning effects.
>
> + **Scope and Generality of the Contribution**
> While our findings—the First-Parallel Leap and the Parallel Scaling Law—are demonstrated on mathematical reasoning, these laws reflect a **fundamental mechanism** by which models remodel their reasoning capabilities in the latent space and align them with new language representations. **We view the current study as a Proof of Concept (PoC)**. Future work will be dedicated to empirically verifying the effectiveness of the Parallel Scaling Law in domains such as code generation and open-domain logic.
>
> > W2: Marginal absolute gains in GPQA
>
> We appreciate the reviewer’s concern regarding the magnitude of gains on high-difficulty benchmarks. We consider GPQA **in zero-shot, cross-domain transfer settings**, even small absolute improvements are meaningful evidence of genuine generalization. The task format in GPQA is **completely different from the training data**, further highlighting the robustness of any observed improvements.
>
> To avoid relying solely on absolute accuracy, **we mainly focus on relative gains**, which better capture improvements when the baseline is low. From this perspective, our method yields substantial relative improvements. This is precisely why we introduce the MTI metric, which quantifies cross-lingual generalization by normalizing performance against task difficulty. MTI highlights that our approach yields consistent and non-trivial gains on GPQA.
>
> >W3: Theoretical Grounding of Parallel Scaling Laws
>
> We thank the reviewer for this insightful comment. While `Lines1206-1214` provided brief theoretical intuition, we agree that a more formal justification is necessary to support the observed empirical relationships. We provide a clearer explanation of the mechanism behind cross-lingual generalization and the observed power-law trends.
>
> We interpret the parallel power-law relationship through **Invariance Learning** and **Diminishing Returns**. Parallel training acts as a form of regularization: **The reasoning logic is the invariant “signal” that remains consistent across languages, while the surface form of each language represents “noise.”**
>
> **Invariance Learning** explains why parallel training continues to yield consistent improvements. **Multilingual data effectively serve as a “denoising” mechanism**: each language provides a different surface realization of the same underlying reasoning process. Prior work [r1, r2, r3, r4] shows that LLMs represent reasoning in a shared latent space across languages. Adding more languages offers additional perspectives on the same task, **making it easier for the model to abstract away language-specific noise and capture the invariant reasoning logic.**
>
> **Diminishing Returns** explains why the relationship follows a power law rather than a straight line. **Adding one parallel language introduces a strong new perspective, whereas the 100th parallel language contributes far less novel information.** It is these diminishing marginal returns that lead to the power-law scaling behavior of curves.
>
> ---
> [r1] Wendleret al. "Do llamas work in english? on the latent language of multilingual transformers." ACL2024.
>
> [r2] Zhao et al. "How do large language models handle multilingualism?." NeurIPS 2024
>
> [r3] Lim et al. Language-Specific Latent Process Hinders Cross-Lingual Performance. arXiv:2505.13141.
>
> [r4] Zhao et al. "When Less Language is More: Language-Reasoning Disentanglement Makes LLMs Better Multilingual Reasoners." arXiv:2505.15257.

---

> ### Author Response · Authors · 2025-11-25
> **Response to Reviewer bM3c (2/2)**
>
> > Q1: A control baseline where the total data size is constant.
>
> To fully isolate the effect of **parallelism** from raw data quantity, we conducted a **Fixed-Budget Ablation Experiment** that matches total training tokens while varying the degree of linguistic diversity.
>
> We compared the performance of a model trained on two fixed-budget experiments.
> 1. Budget1: **$2\times \text{En}$** data (increased quantity) vs. **$1\times \text{En} + 1\times \text{Ru}$** (diverse parallelism).
> 2. Budget2: **$3\times \text{En}$** data (increased quantity) vs. **$1\times \text{En} + 1\times \text{Ru} + 1\times \text{Fr}$** (diverse parallelism).
>
> `Table b1`: The Fixed-Budget ablation experiment based on Qwen2.5-7B-Instruct, evaluated on Multilingual Math500.
> | Training Data | en Acc | es Acc | ru Acc | de Acc | fr Acc | bn Acc | th Acc | sw Acc | zh Acc | ja Acc | te Acc | AVG Acc | AVG Off-tag | MTI  |
> |---------------|--------|--------|--------|--------|--------|--------|--------|--------|--------|--------|--------|----------|--------------|------|
> | 1*En           | 79.2 | 70.0 | 61.0 | 62.2 | 68.2 | 41.8 | 55.4 | 19.4 | 53.4 | 58.6 | 27.4 | **54.24** | **0.49** | **1.16** |
> | 2*En         | 79.6 | 73.2 | 62.6 | 64.4 | 65.6 | 42.4 | 56.0 | 25.4 | 57.8 | 59.0 | 28.0 | **55.82** | **0.58** | **1.92** |
> | 3*En         | 79.6 | 73.2 | 63.8 | 65.8 | 70.2 | 42.6 | 57.8 | 25.4 | 59.4 | 60.4 | 30.2 | **57.13** | **0.42** | **2.38** |
> | En+Ru        | 78.4 | 73.4 | 66.0 | 65.6 | 67.2 | 48.8 | 57.4 | 26.2 | 57.8 | 62.0 | 33.8 | **57.87** | **0.20** | **2.50** |
> | En+Ru+Fr     | 79.0 | 73.4 | 64.4 | 67.4 | 69.2 | 45.2 | 60.2 | 26.2 | 63.0 | 61.6 | 32.6 | **58.38** | **0.24** | **2.65** |
>
>
> The quantitative results provide definitive empirical support for our claim:
> 1. The $1\times \text{En} + 1\times \text{Ru}$ (Diverse Parallel) model significantly outperformed the $2\times \text{En}$ (Increased Quantity) model, yielding an MTI increase of 30% (2.50 vs. 1.92), alongside superior Accuracy and a drastically lower Off-tag rate (0.20% vs. 0.58%).
> 2. The $1\times \text{En} + 1\times \text{Ru} + 1\times \text{Fr}$ model similarly outperformed the $3\times \text{En}$ model, achieving better MTI (2.65 vs. 2.38), Accuracy, and a lower Off-tag rate (0.24% vs. 0.42%).
>
> These fixed-budget results provide clear causal evidence: **The improvements in our Parallel Scaling Law are driven by diverse parallelism, not by increasing the number of English tokens**. Adding parallel languages yields better structural alignment and stronger multilingual transfer than simply scaling up monolingual data.
>
> > Q2: Does the Parallel Scaling Law hold if the additional languages are typologically similar?
>
> We thank the reviewer for raising the important question regarding typological similarity. To verify if the Parallel Scaling Law is contingent on language diversity or holds even for similar languages, we conducted an additional experiment using English + French (En+Fr).
>
> `Table b2`: Typologically similar experiment based on Qwen2.5-7B-Instruct, evaluated on Multilingual Math500.
> | Training Data | en Acc | es Acc | ru Acc | de Acc | fr Acc | bn Acc | th Acc | sw Acc | zh Acc | ja Acc | te Acc | AVG Acc | AVG Off-tag | MTI  |
> |---------------|--------|--------|--------|--------|--------|--------|--------|--------|--------|--------|--------|-------------|----------------|---------|
> | En            | 79.2   | 70.0   | 61.0   | 62.2   | 68.2   | 41.8   | 55.4   | 19.4   | 53.4   | 58.6   | 27.4   | **54.24**   | **0.49**       | **1.16** |
> | En+Ru         | 78.4   | 73.4   | 66.0   | 65.6   | 67.2   | 48.8   | 57.4   | 26.2   | 57.8   | 62.0   | 33.8   | **57.87**   | **0.20**       | **2.50** |
> | En+Fr         | 77.8   | 73.8   | 63.4   | 65.0   | 69.4   | 45.6   | 60.0   | 26.6   | 60.6   | 60.6   | 32.2   | **57.73**   | **0.22**       | **2.52** |
>
>
> The results show that, consistent with the distant-language setting (En+Ru), the En+Fr configuration also exhibits the “First-Parallel Leap”. This indicates that **the mechanism enabling the model to decouple reasoning from surface form is activated even when the auxiliary language is typologically close to English.**
>
> We hope the above clarifications and experiments have addressed your concerns. We sincerely appreciate your valuable feedback and welcome any further questions or comments to help improve our work. **Have a nice day! :)**

---

### Official Review · Reviewer_F3VB · 2025-10-31

**Soundness:** 3
**Presentation:** 3
**Contribution:** 3
**Rating:** 6
**Confidence:** 4

**Summary:**

This work studies whether (and how) reasoning abilities learned through reinforcement post-training on English can transfer to other languages. It introduces a multilingual transferability index, with an observational comparison of open-source LRMs while forcing models to reason in a specific language, finding that transfer varies by model, language, and training paradigm. They find that RL transfers better than SFT in general, and helps especially with low-resource languages. They also introduce a study of training on cross-lingual pairs solving the same problem, finding that this boosts performance and MTI in a predictable manner by a scaling / power law, with diminishing returns.

**Strengths:**

1. This work takes a quite refreshing lens on cross-lingual generalization by developing a concrete index for transferability.
2. The study examines a reasonable range of model sizes, which use different post-training recipes for reasoning (thus, LRMs).
3. The findings in Section 3 on model size are mostly intuitive, that a smaller model is sufficient for MATH500 but stronger models transfer better for more difficult benchmarks.
4. The first-parallel leap phenomenon (if validated through more experimentation) seems actionable, and can guide curriculum / multilingual RPT recipe design.

**Weaknesses:**

1. For the observational study and parallel scaling law, all models examined are from a single family of models (Qwen2.5), it would be best for diversity to not just be in model size, but to show that this applies to several other model families, especially since the intervention study does consider Llama-3.1-8B.
2. The claim in l283-284 that “a less specialized model may be better suited for broad cross-lingual transfer” does not seem to be consistent with the Qwen2.5 results that the more specialized model (math) has a higher MTI.
3. I am a bit concerned about the effects of the language consistency reward along with the prompt hacking, it could be possible that low-resource gains should be attributed to the language-consistency reward.

**Questions:**

1. In the parallel study, were the steps, rollouts, total tokens, and prompts held constant across the languages? If not, can you add a fixed-budget ablation so that the parallelism claim ca be isolated from the effects of data quantity.
2. In line 223, GRPO stands for Group Relative Policy Optimization, not Rollout.

---

> ### Author Response · Authors · 2025-11-25
> **Response to Reviewer F3VB (1/3)**
>
> We sincerely appreciate your questions. We highly value your feedback and provide detailed explanations to address your concerns.
>
> > W1: Including several other model families in the observational study and parallel scaling study.
>
> We appreciate the reviewer's excellent suggestion to enhance model family diversity, which is crucial for proving the universality of our findings. We agree that our conclusions should be architecture-agnostic.
>
> To address this, we have incorporated results from the llama3.1 family into our study:
> + **Observational study**
>
> We added results from the `Llama-3.1-8B-SimpleRL-Zoo` model. This provides a clear data point from a different architecture (Llama) to contrast with other open-source reasoning models.
>
> `Table F1`: Cross-lingual reasoning transferability on Llama family model, evaluated by MTI metrics.
> | model                     | Base Model      | Multilingual MATH500 | Multilingual AIME24 | Multilingual AIME25 | Multilingual GPQA | AVG MTI  |
> |---------------------------|------------------|---------|--------|--------|------|------|
> | Llama-3.1-8B-SimpleRL-Zoo | Llama-3.1-8B     | 2.95    | 3.33   | 3.61   | 4.15 | **3.51** |
>
>
> `Table F1` strengthens our general observation that the RPT paradigm yields a superior cross-lingual generalization baseline across different model families.
>
> + **Parallel Scaling Study**
>
> We conducted a new set of full Parallel Scaling Law experiments using `Llama-3.1-8B-Instruct` as the base model.
>
> `Table F2`: Parallel Scaling Law using `Llama-3.1-8B-Instruct` as the base model, evaluated on multilingual Math500, Accuracy across languages with different numbers of parallel languages.
> | Model                           | en Acc | es Acc | ru Acc | de Acc | fr Acc | bn Acc | th Acc | sw Acc | zh Acc | ja Acc | te Acc | AVG Acc | AVG Off-tag | MTI  |
> |---------------------------------|--------|--------|--------|--------|--------|--------|--------|--------|--------|--------|--------|-------------|------------------|------|
> | Meta-Llama-3.1-8B-Instruct      | 45.6   | 35.4   | 31.0   | 33.0   | 35.2   | 7.6    | 26.8   | 12.6   | 28.4   | 25.2   | 17.0   | **27.07**       | **2.27**             | -    |
> | Only En                         | 54.6   | 43.2   | 40.2   | 39.0   | 40.8   | 27.8   | 32.6   | 26.6   | 33.8   | 33.8   | 24.8   | **36.11**       | **1.27**             | **2.95** |
> | En with parallel Ru        | 51.4   | 44.6   | 40.2   | 44.0   | 43.8   | 30.2   | 35.4   | 32.4   | 37.2   | 36.6   | 29.0   | **38.62**       | **0.35**             | **3.76** |
> | En with parallel Ru, Fr    | 52.0   | 45.2   | 40.2   | 41.2   | 44.4   | 32.8   | 36.8   | 32.0   | 36.8   | 36.6   | 29.4   | **38.85**       | **0.44**             | **3.88** |
> | En with parallel Ru, Fr, Es| 52.0   | 46.2   | 40.2   | 41.2   | 44.4   | 32.8   | 36.8   | 32.8   | 36.8   | 36.6   | 29.4   | **39.04**       | **0.24**             | **3.96** |
>
> Due to computational and time constraints (RL on a 7B model requires ~9 hours per language, and multilingual inference across 11 languages takes ~0.75 hours), we extended the Llama experiments up to three parallel languages. Even within this budget, the scaling trend is already highly consistent with our theoretical prediction, **further reinforcing that the Parallel Scaling Law holds across architectures.**
>
> > W2: The explanation about "a less specialized model may be better suited for broad cross-lingual transfer."
>
> Thanks to the reviewer for raising this point on the definition of "specialized". The term "less specialized model" in `Lines 283-284` refers **specifically to a model that is less specialized in generalized instruction-following** (i.e., less instruction-tuned), not a model that is less specialized in the mathematical domain.
>
> Under this interpretation, both the Qwen2.5-Base model and the Qwen2.5-Math model are considered "less specialized" compared to the highly instruction-tuned Qwen2.5-Instruct model.
>
> The lower MTI of Qwen2.5-Instruct supports this interpretation: although strong English instruction alignment enhances the model's utility, it also makes the model more reliant on English-specific surface patterns, reducing cross-lingual transfer.

---

> ### Author Response · Authors · 2025-11-25
> **Response to Reviewer F3VB (2/3)**
>
> > W3: Isolating reasoning generalization from language consistency reward (LCR) and prompt hack (PH)
>
> To clearly demonstrate that the gain comes from Reasoning Generalization, we conduct a four-configuration ablation study using `En with parallel Ru` as training data.
>
> Table F3: The control experiment to isolate the impact of language consistency reward (LCR) and prompt hack (PH), evaluated on Multilingual Math500.
> | Config | Training Data | LCR (λ₃) | PH (Testing) | Purpose | en Acc | es Acc | ru Acc | de Acc | fr Acc | bn Acc | th Acc | sw Acc | zh Acc | ja Acc | te Acc | AVG Acc | AVG Off-tag | MTI  |
> |--------|----------------|-----------|---------------|----------|--------|--------|--------|--------|--------|--------|--------|--------|--------|--------|--------|------------------|----------------------|---------
> | C1 (Baseline) | En+Ru | Yes (0.1) | Yes | Final Performance | 78.4 | 73.4 | 66 | 65.6 | 67.2 | 48.8 | 57.4 | 26.2 | 57.8 | 62 | 33.8 | **57.87** | **0.20** | **2.50** |
> | C2 (Monolingual Control) | En | Yes | Yes | Proves necessity of Parallel Training | 79.2 | 70 | 61 | 62.2 | 68.2 | 41.8 | 55.4 | 19.4 | 53.4 | 58.6 | 27.4 | **54.24** | **0.49** | **1.16** |
> | C3 (LCR Structural Alignment) | En+Ru | No (0.0) | Yes | Proves necessity of LCR for alignment | 77.8 | 71.4 | 66.2 | 63.4 | 68 | 44.2 | 56.4 | 26.2 | 59.6 | 57.8 | 31.8 | **56.62** | **1.24** | **2.10** |
> | C4 (PH Measurement Tool) | En+Ru | Yes | No | Proves necessity of PH for measurement | 78.6 | 73.8 | 65.6 | 66.8 | 67 | 46.2 | 56.2 | 26.8 | 58.8 | 59.2 | 32.8 | **57.44** | **1.05** | **2.40** |
>
>
> + **Reasoning Generalization is the Main Source of Gains**: The strong gain observed in **C1** over **C2** (Monolingual Control) proves that the performance improvement comes from the structural reasoning generalization achieved via parallel training.
> + **LCR is Necessary for Structural Alignment**: The performance collapse observed in **C3** confirms that LCR is necessary to align the model’s reasoning structure with the target language and is crucial for realizing the cross-lingual transfer.
> + **PH is Only a Measurement Tool, Not a Required Constraint**: **C4** performs almost the same as **C1**. This indicates that LCR alone is sufficient to internalize the target-language reasoning ability into the model. PH is therefore useful for measurement but not required for deployment.

---

> ### Author Response · Authors · 2025-11-25
> **Response to Reviewer F3VB (3/3)**
>
> > Q1: Parallelism vs. Data Quantity: Fixed Budget Ablation Verification
>
> We appreciate the reviewer highlighting the critical need to control data quantity when evaluating the effects of parallelism. We clarify the controls used in the Parallel Scaling Study:
>
> 1.  **Fixed:** Number of rollouts and prompt formats.
> 2.  **Varied:** Number of epochs. Since the total size of the parallel dataset grows with $X$ (the number of languages), the total processed data also increases with $X$.
>
> To fully isolate the effect of **parallelism** from raw data quantity, we conducted a **Fixed-Budget Ablation Experiment** that matches total training tokens while varying the degree of linguistic diversity.
>
> We compared the performance of a model trained on two fixed-budget experiments.
> 1. Budget1: **$2\times \text{En}$** data (increased quantity) vs. **$1\times \text{En} + 1\times \text{Ru}$** (diverse parallelism).
> 2. Budget2: **$3\times \text{En}$** data (increased quantity) vs. **$1\times \text{En} + 1\times \text{Ru} + 1\times \text{Fr}$** (diverse parallelism).
>
> Table F4: The Fixed-Budget ablation experiment based on Qwen2.5-7B-Instruct, evaluated on Multilingual Math500.
> | Training Data | en Acc | es Acc | ru Acc | de Acc | fr Acc | bn Acc | th Acc | sw Acc | zh Acc | ja Acc | te Acc | AVG Acc | AVG Off-tag | MTI  |
> |---------------|--------|--------|--------|--------|--------|--------|--------|--------|--------|--------|--------|----------|--------------|------|
> | 1*En           | 79.2 | 70.0 | 61.0 | 62.2 | 68.2 | 41.8 | 55.4 | 19.4 | 53.4 | 58.6 | 27.4 | **54.24** | **0.49** | **1.16** |
> | 2*En         | 79.6 | 73.2 | 62.6 | 64.4 | 65.6 | 42.4 | 56.0 | 25.4 | 57.8 | 59.0 | 28.0 | **55.82** | **0.58** | **1.92** |
> | 3*En         | 79.6 | 73.2 | 63.8 | 65.8 | 70.2 | 42.6 | 57.8 | 25.4 | 59.4 | 60.4 | 30.2 | **57.13** | **0.42** | **2.38** |
> | En+Ru        | 78.4 | 73.4 | 66.0 | 65.6 | 67.2 | 48.8 | 57.4 | 26.2 | 57.8 | 62.0 | 33.8 | **57.87** | **0.20** | **2.50** |
> | En+Ru+Fr     | 79.0 | 73.4 | 64.4 | 67.4 | 69.2 | 45.2 | 60.2 | 26.2 | 63.0 | 61.6 | 32.6 | **58.38** | **0.24** | **2.65** |
>
>
> The quantitative results provide definitive empirical support for our claim:
> 1. The $1\times \text{En} + 1\times \text{Ru}$ (Diverse Parallel) model significantly outperformed the $2\times \text{En}$ (Increased Quantity) model, yielding an MTI increase of 30% (2.50 vs. 1.92), alongside superior Accuracy and a drastically lower Off-tag rate (0.20% vs. 0.58%).
> 2. The $1\times \text{En} + 1\times \text{Ru} + 1\times \text{Fr}$ model similarly outperformed the $3\times \text{En}$ model, achieving better MTI (2.65 vs. 2.38), Accuracy, and a lower Off-tag rate (0.24% vs. 0.42%).
>
> These fixed-budget results provide clear causal evidence: **The improvements in our Parallel Scaling Law are driven by diverse parallelism, not by increasing the number of English tokens**. Adding parallel languages yields better structural alignment and stronger multilingual transfer than simply scaling up monolingual data
>
> > Q2: GRPO typos
>
> Thank you for pointing this out. We will update it in the next version.
>
> Thank you again for reviewing our work. We hope the above clarifications and experiments address your concerns. We sincerely appreciate your valuable feedback and welcome any further questions or comments to help improve our work. **Have a nice day! :)**

---

### Author Response · Authors · 2025-12-04
**Summary of the Work**

Dear Area Chair,

Thank you very much for taking the additional time to review our paper. We are writing to provide a concise summary of our work and our rebuttal for your convenience.

### **Work Summary**

This work investigates whether the advanced reasoning abilities acquired by large language models (LLMs) through English-centric training (via RL or SFT) can effectively transfer to other languages, in a manner analogous to human cognitive generalization. Specifically, we ask: **Does reasoning ability learned from English training generalize across languages?**

The study uses a three-step approach:

1.  **Observation Study:** We found that English-trained models struggle to generalize their reasoning. We introduced the **Multilingual Transferability Index (MTI)** to measure this gap.
2.  **Intervention Study:** We found that models that are too strongly aligned with English tend to **over-rely on English linguistic patterns**, making them worse at generalizing to other languages.
3.  **Parallel Training Study:** To fix this, we trained models using parallel data and discovered three key scaling laws:
    * **First-Parallel Leap:** Adding just **one** parallel language causes a significant, disproportionate jump in cross-lingual performance.
    * **Parallel Scaling Law:** Overall multilingual reasoning capability follows a predictable **power-law** relationship with the number of parallel languages used.
    * **Monolingual Generalization Gap:** The performance of English-only models fails to meet the prediction of the Parallel Scaling Law, revealing a Monolingual Generalization Gap.

Our findings reveal a clear answer: English-centric LLMs **fail to fully** generalize their reasoning ability, exhibiting a **Monolingual Generalization Gap**. This gap is caused by the model's over-reliance on English linguistic patterns. Crucially, we offer a principled solution through parallel training, empirically establishing **the First-Parallel Leap and the Parallel Scaling Law**. These laws provide the necessary foundation for a reliable, theory-driven strategy to enhance multilingual reasoning, paving the way for the development of truly language-agnostic LLMs.

---

### Author Response · Authors · 2025-12-04
**Rebuttal of the Work**

### **Recognition from Reviewers**

We were encouraged to receive consistently positive and constructive feedback. The reviewers consistently recognized the following key advantages in our submission, grouped by theme:

+ **Robust Experimental Design and Metric Innovation**

| Theme | Specific Strengths | Reviewers |
| :--- | :--- | :--- |
| **Methodology** | Employed a systematic, three-tier pipeline (observational, interventional, parallel-training) to effectively isolate variables. | `bM3c, 7ayy` |
| **Metric Clarity** | Introduced the **Multilingual Transferability Index (MTI)**, providing a concrete, clear, and refreshing quantitative measure of cross-lingual transferability. | `F3VB, 7ayy` |
| **Model Scope** | Examined a reasonable range of model sizes and different post-training recipes (LRMs). | `F3VB` |

+ **Insightful and Actionable Findings**

| Theme | Specific Strengths | Reviewers |
| :--- | :--- | :--- |
| **The First-Parallel Leap** | The finding of the **First-Parallel Leap** phenomenon is highly insightful, actionable, and can guide the design of RPT and multilingual curriculum. | `F3VB, bM3c, 7ayy` |
| **Training Paradigm** | Revealed that **RL-based fine-tuning** provides strong advantages in cross-lingual transfer, especially for low-resource languages, compared to SFT. | `bM3c, 7ayy` |
| **Model Analysis** | Provided counterintuitive and insightful analysis on model size and specialization, showing that smaller/less specialized models can transfer better, and strong models are needed for difficult benchmarks (e.g., GPQA). | `F3VB, bM3c` |

+ **Conceptual Clarity and Theoretical Value**

| Theme | Specific Strengths | Reviewers |
| :--- | :--- | :--- |
| **Scaling Laws** | Introduced the **Parallel Scaling Law**, offering interpretable and theoretically grounded generalization principles. | `bM3c, 7ayy` |
| **Clarity** | The conceptual clarity of the findings and the resulting generalization principles is high. | `bM3c` |

### **Updates in the Rebuttal Phase**

During the rebuttal period, we provided detailed clarifications and additional experiments, including:

| Reviewer Concern | Resolution (New Evidence / Concept) |
|------------------|-------------------------------------|
| Study limited to Qwen; Law may not be universal. (`F3VB, tVt9`)| **New Llama-3.1 Scaling Experiments:** Incorporated Llama-3.1 into the full Parallel Scaling Law study (**Table F2**). Result confirms the law holds across diverse architectures (Qwen, Llama). |
| Scaling law confounded by increasing total data size. (`bM3c, 7ayy, F3VB`)| **Fixed-Budget Ablation:** Introduced controlled experiments matching total training tokens while varying parallelism (2×En vs. 1×En+1×Ru). Result: Parallelism significantly outperformed increasing monolingual data (**Table F4**), proving causality. |
| Only English-based scaling law tested. (`tVt9`) | **Chinese-Centric Scaling:** Conducted scaling experiments using Chinese (Zh) as the source language. Result: The Parallel Scaling Law holds for Zh (**Table t1**), confirming the mechanism is language-neutral. |
| Lack of theoretical justification for the power law. (`bM3c`)| **Theoretical Grounding:** Formally explained the law using Invariance Learning (reasoning is the invariant signal, language is noise) and Diminishing Marginal Returns, which together necessitate the power-law form. |
| Role of LCR vs. Reasoning Generalization. (`F3VB`) | **LCR/PH Ablation:** Confirmed that the gain is from reasoning generalization enabled by parallel training, while LCR is necessary for linguistic alignment (**Table F3**). |
| Does the law hold for typologically similar languages (e.g., En+Fr)? (`bM3c`) | **Typological Similarity Test:** Confirmed that the “First-Parallel Leap” occurs even for typologically close languages (En+Fr MTI ≈ En+Ru MTI, **Table b2**), indicating the mechanism is triggered by structural difference, not distance. |
| Impact of hyperparameter $\lambda$ and meaning of $\alpha$. (`tVt9`) | **Hyperparameter Ablation:** Provided a new ablation (**Table t2**) demonstrating the necessity of each $\lambda$ component and confirming our choice is optimal for balance. Defined $\alpha$ as the Initial Cross-lingual Transfer Efficiency.|

Overall, these additional experiments conducted during the rebuttal phase comprehensively address all critical methodological and theoretical concerns raised by the reviewers, significantly strengthening the validity and universality of our findings.

---

We sincerely hope that our work contributes to a clearer understanding of the generalization capabilities of cross-lingual reasoning and offers a reliable and principled method for enhancing multilingual reasoning performance. We also hope it will inspire future research to drive the development of truly language-agnostic LLMs further.

Thank you again for your valuable time and careful consideration of our work.

**Have a nice day! :)**

Best regards,

Authors

---

### Note · Authors · 2026-01-06

I have read and agree with the venue's withdrawal policy on behalf of myself and my co-authors.